# SubTab: Subsetting Features of Tabular Data for Self-Supervised Representation Learning

**Talip Uçar, Ehsan Hajiramezanali, Lindsay Edwards**

Respiratory and Immunology, R&D, AstraZeneca
{talip.ucar, ehsan.hajiramezanali, lindsay.edwards}@astrazeneca.com

## Abstract

Self-supervised learning has been shown to be very effective in learning useful representations, and yet much of the success is achieved in data types such as images, audio, and text. The success is mainly enabled by taking advantage of spatial, temporal, or semantic structure in the data through augmentation. However, such structure may not exist in tabular datasets commonly used in fields such as healthcare, making it difficult to design an effective augmentation method, and hindering a similar progress in tabular data setting. In this paper, we introduce a new framework, **Sub**setting features of **Tab**ular data (SubTab), that turns the task of learning from tabular data into a multi-view representation learning problem by dividing the input features to multiple subsets. We argue that reconstructing the data from the subset of its features rather than its corrupted version in an autoencoder setting can better capture its underlying latent representation. In this framework, the joint representation can be expressed as the aggregate of latent variables of the subsets at test time, which we refer to as *collaborative inference*. Our experiments show that the SubTab achieves the state of the art (SOTA) performance of 98.31% on MNIST in tabular setting, on par with CNN-based SOTA models, and surpasses existing baselines on three other real-world datasets by a significant margin.

## 1 Introduction

In recent years, the self-supervised learning has successfully been used to learn meaningful representations of the data in natural language processing [34, 41, 11, 28, 10, 21, 9]. A similar success has been achieved in image and audio domains [7, 15, 37, 5, 17, 13, 8]. This progress is mainly enabled by taking advantage of spatial, semantic, or temporal structure in the data through data augmentation [7], pretext task generation [11] and using inductive biases through architectural choices (e.g. CNN for images). However, these methods can be less effective in the lack of such structures and biases in the tabular data commonly used in many fields such as healthcare, advertisement, finance, and law. And some augmentation methods such as cropping, rotation, color transformation etc. are domain specific, and not suitable for tabular setting. The difficulty in designing similarly effective methods tailored for tabular data is one of the reasons why self-supervised learning is under-studied in this domain [46].

The most common approach in tabular data is to corrupt data through adding noise [43]. An autoencoder maps corrupted examples of data to a latent space, from which it maps back to uncorrupted data. Through this process, it learns a representation robust to the noise in the input. This approach may not be as effective since it treats all features equally as if features are equally informative. However, perturbing uninformative features may not result in the intended goal of the corruption. A recent work takes advantage of self-supervised learning in tabular data setting by introducing a pretext task [46], in which a de-noising autoencoder with a classifier attached to representation layer is trained on

35th Conference on Neural Information Processing Systems (NeurIPS 2021).

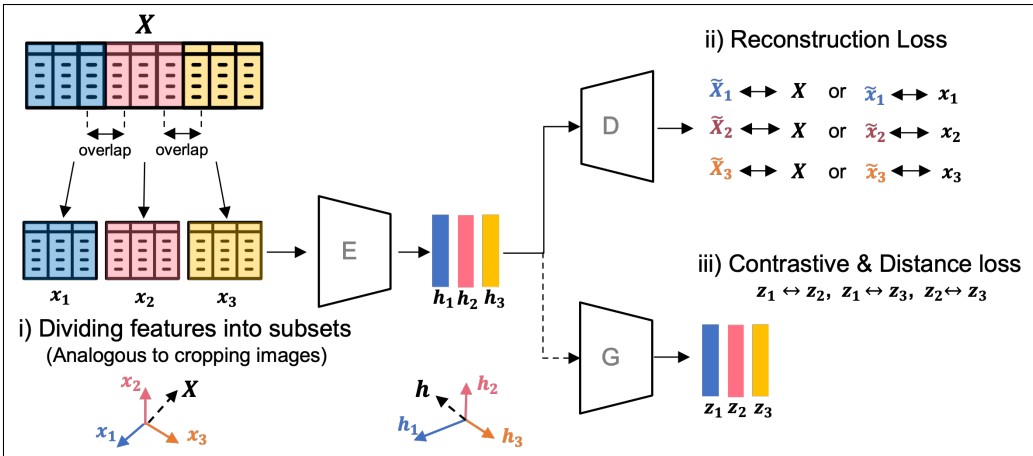

Figure 1: SubTab framework: i) Dividing the features into subsets (similar to feature bagging, or cropping images), ii) Reconstruction of either subsets of features ($\tilde{x}_1, \tilde{x}_2, \tilde{x}_3$), or complete feature space ($\tilde{X}_1, \tilde{X}_2, \tilde{X}_3$), which are used to compute reconstruction loss. iii) Generating projections used to compute contrastive and distance loss. $E \equiv Encoder, D \equiv Decoder, G \equiv Projection$.

corrupted data. The classifier's task is to predict the location of corrupted features. However, this framework still relies on noisy data in the input. Additionally, training a classifier on an imbalanced binary mask for a high-dimensional data may not be ideal to learn meaningful representations.

In this work, we turn the problem of learning representation from a single-view of the data into the one learnt from its multiple views by dividing the features into subsets, akin to cropping in image domain or feature bagging in ensemble learning, to generate different views of the data. Each subset can be considered a different view. We show that reconstructing data from the subset of its features forces the encoder to learn better representation than the ones learned through the existing methods such as adding noise. We train our model in a self-supervised setting and evaluate it on downstream tasks such as classification, and clustering. We use five different datasets; MNIST in tabular format, the cancer genome atlas (TCGA) [42], human gut metagen-omic samples of obesity cohorts (Obesity) [36, 26], UCI adult income (Income) [24], and UCI BlogFeedback (Blog) [4].

SubTab can: i) construct a better representation by using the aggregate of the representation of the subsets, a process that we refer as *collaborative inference* ii) discover the regions of informative features by measuring predictive power of each subset, which is useful especially in high-dimensional data iii) do training and inference in the presence of missing features by ignoring corresponding subsets and iv) use smaller models by reducing input dimension, making it less prone to overfitting.

## 2 Method

The augmentation methods such as adding noise, rotation, cropping etc. are commonly used in image domain. Among them, the cropping is shown to be the most effective technique [7]. Inspired from this insight, we propose subsetting features of tabular data.

Figure 1 presents SubTab framework, in which we have an encoder (E), a decoder (D), and an optional projection (G). For the purpose of this paper, we will refer $h$ as latent, or representation, $z$ as projection, $\tilde{x}$, and $\tilde{X}$ as the reconstruction of subset, and whole data respectively. Small letters are associated with subsets while capital latters are associated the whole set of features. Moreover, throughout this work, when we say that a representation is "good", we refer to its performance in a classification task using a linear model.

In SubTab framework, we divide tabular data to multiple subsets. Neighbouring subsets can have overlapping regions, defined as a percentage of a dimension of the subset. Each of the subsets is fed to the same encoder (i.e. parameter sharing) to get their corresponding latent representation. A shared decoder is used to reconstruct either the subset fed to the encoder, or full tabular data (i.e. reconstructing all features from the subset of features). We chose the latter in our experiments since

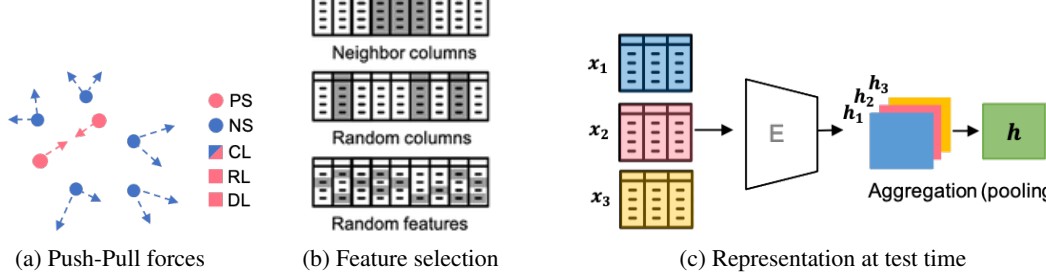

|  (a) Push-Pull forces | (b) Feature selection | (c) Representation at test time |

Figure 2: a) Push-Pull forces applied by each loss. PS / NS : Positive/Negative sample; CL/RL/DL: Contrastive, Reconstruction, Distance losses b) Column or feature selection strategies for adding noise to each subset. **Top:** Selecting a block of neighbouring columns; **Middle:** Selecting columns randomly; **Bottom:** Selecting random features per row c) Latent variables from each subset is aggregated at test time. The mean (default), sum, max, or min aggregation can be used.

it is more effective in learning good representations. We should also note that, in the latter case, the autoencoder cannot learn the identity, eliminating the constraint on the dimension of the bottleneck (i.e. representation). We compute one reconstruction loss term per subset.

Moreover, we can optionally add contrastive loss to our objective by using all combination of pairs of projections from all subsets. For example, given three subsets as in Figure 1, there are three combinations of two: $\binom{n}{k} = \binom{3}{2} = \frac{3!}{2!(1)!} = 3$ . For four subsets, it would be 6 pairs of combination, and so on. We can add one more loss term, referred as distance loss, to reduce the distance between the pairs of projections of the subsets by using a loss function such as mean squared error (MSE). All three loss terms apply a pulling force on positive samples while contrastive loss also applies a push force between positive and negative samples as shown in Figure 2a.

Once the dataset is divided into subsets in data preparation step, a process that is similar to feature bagging in ensemble learning, their location is fixed. Thus, *we don't change the relative order of features in a subset* during training since standard neural network architectures are not permutation invariant. This is to ensure that same features are fed to the same input units of neural network. However, our method can be extended to permutation invariant setting as a next step.

## 2.1 Strategies for adding noise

Our framework is complementary to other augmentation techniques used in tabular data setting. Thus, we experimented with adding noise to randomly selected entries in each subset by using three types of noise: i) adding Gaussian noise, $\mathcal{N}(0, \sigma^2)$, ii) overwriting the value of a selected entry with another value randomly sampled from the same column, referred as swap-noise, iii) zeroing-out randomly selected entries, referred as zero-out noise.

Moreover, we use three different strategies when selecting the features to add noise to, as shown in Figure 2b: i) a random block of neighboring columns (NC), ii) random columns (RC) iii) random features per each sample (RF). To add noise, we create a binomial mask, $m$, and a noise matrix, $\epsilon$, with same shape as the subset, in which the entries of the mask is assigned to 1 with probability $p$, and to 0 otherwise. The corrupted version, $x_{1c}$, of subset $x_1$ is generated as following:

$$x_{1c} = (1 - m) \odot x_1 + m \odot \epsilon \tag{1}$$

## 2.2 Training

Our objective function is:

$$\mathcal{L}_t = \mathcal{L}_r + \mathcal{L}_c + \mathcal{L}_d, \tag{2}$$

where $\mathcal{L}_t$, $\mathcal{L}_r$, $\mathcal{L}_c$ and $\mathcal{L}_d$ are total, reconstruction, contrastive, and distance losses, respectively.

**i) Reconstruction loss:** Given a subset, denoted by $x_k$, we can reconstruct either the same subset, $\tilde{x}_k$ or the entire feature space $\tilde{X}_k$. Then, we can compute the reconstruction loss for $k^{th}$ subset by

computing mean squared error using either $(\boldsymbol{x_k}, \tilde{\boldsymbol{x}}_{\boldsymbol{k}})$, or $(\boldsymbol{X}, \tilde{\boldsymbol{X}}_{\boldsymbol{k}})$ pair as shown in Figure 1. We chose the latter since it was more effective. Overall reconstruction loss:

$$\mathcal{L}_r = \frac{1}{K} \sum_{k=1}^{K} s_k, \text{ where } s_k = \frac{1}{N} \sum_{i=1}^{N} \left( \boldsymbol{X}^{(i)} - \tilde{\boldsymbol{X}}_{\boldsymbol{k}}^{(i)} \right)^2 \tag{3}$$

where $K$ is the total number of subsets, $N$ is the size of the batch, $s_k$ is the reconstruction loss for $k^{th}$ subset, and $\mathcal{L}_r$ is the average of reconstruction loss over all subsets.

**ii) Contrastive loss:** If the dataset is rich in the number of classes such that chances of sampling negative samples are high, we can use a projection network (G) to get projections, $\boldsymbol{z}'s$, of representations, $\boldsymbol{h}'s$. Samples at the same rows of two subsets, $\boldsymbol{z_1}$ and $\boldsymbol{z_2}$, can be considered as positive pairs while remaining rows in the subsets can be considered as negative to those samples. This allows us to compute the contrastive loss for each pair of projections using a loss function such as the normalized temperature-scaled cross entropy loss (NT-Xent) [7]. For three subsets, $\{\boldsymbol{x_1}, \boldsymbol{x_2}, \boldsymbol{x_3}\}$, we can compute such a loss for every pair $\{\boldsymbol{z_a}, \boldsymbol{z_b}\}$ of total three pairs from the set $S = \{\{\boldsymbol{z_1}, \boldsymbol{z_2}\}, \{\boldsymbol{z_1}, \boldsymbol{z_3}\}, \{\boldsymbol{z_2}, \boldsymbol{z_3}\}\}$. Overall contrastive loss is:

$$\mathcal{L}_c = \frac{1}{J} \sum_{\{\boldsymbol{z_a}, \boldsymbol{z_b}\} \in S} p(\boldsymbol{z_a}, \boldsymbol{z_b}), \text{ where } p(\boldsymbol{z_a}, \boldsymbol{z_b}) = \frac{1}{2N} \sum_{i=1}^{N} \left[ l(\boldsymbol{z_a}^{(i)}, \boldsymbol{z_b}^{(i)}) + l(\boldsymbol{z_b}^{(i)}, \boldsymbol{z_a}^{(i)}) \right] \tag{4}$$

$$l(\boldsymbol{z_a}^{(i)}, \boldsymbol{z_b}^{(i)}) = -\log \frac{\exp(sim(\boldsymbol{z_a}^{(i)}, \boldsymbol{z_b}^{(i)})/\tau)}{\sum_{k=1}^{N} \mathbb{1}_{k \neq i} \exp(sim(\boldsymbol{z_a}^{(i)}, \boldsymbol{z_b}^{(k)})/\tau)} \tag{5}$$

where $J$ is the total number of pairs in set $S$, $p(\boldsymbol{z_a}, \boldsymbol{z_b})$ is total contrastive loss for a pair of projection $\{\boldsymbol{z_a}, \boldsymbol{z_b}\}$, $l(\boldsymbol{z_a}^{(i)}, \boldsymbol{z_b}^{(i)})$ is the loss function for a corresponding positive pairs of examples $(\boldsymbol{z_a}^{(i)}, \boldsymbol{z_b}^{(i)})$ in subsets $\{\boldsymbol{z_a}, \boldsymbol{z_b}\}$, and $\mathcal{L}_c$ is the average of contrastive loss over all pairs.

**iii) Distance loss:** We can also add mean-squared error (MSE) loss for pairs of projections of subsets since the corresponding samples in subsets should be close to each other. Hence, we can compute an overall MSE loss as:

$$\mathcal{L}_d = \frac{1}{J} \sum_{\{\boldsymbol{z_a}, \boldsymbol{z_b}\} \in S} p(\boldsymbol{z_a}, \boldsymbol{z_b}), \text{ where } p(\boldsymbol{z_a}, \boldsymbol{z_b}) = \frac{1}{N} \sum_{i=1}^{N} \left( \boldsymbol{z_a}^{(i)} - \boldsymbol{z_b}^{(i)} \right)^2 \tag{6}$$

The pseudocode of algorithm can be found in Algorithm 1 in the Appendix. We should note that both $\mathcal{L}_c$ and $\mathcal{L}_d$ in equation (2) are optional, and we used them only in some experiments.

## 2.3 Test time

At test time, we feed the subsets of test set to the encoder, and get the aggregate of the representations of all available subsets as shown in Figure 2c. Please note that we can use mean, sum, min, max, or any other aggregation method to get joint representation, which is analogous to pooling in Computer Vision, or the aggregation of neighbouring nodes in graph convolutional networks [23]. We used mean aggregation in all our experiments, but did compare different aggregation methods in Appendix F.4. Our experiments show that we can use the representations of only one, or few subsets and still achieve a good performance at test time. For example, we could use only $\boldsymbol{h_1}$, or aggregate of $\boldsymbol{h_1}$ and $\boldsymbol{h_2}$ rather than aggregating over all subsets $(\boldsymbol{h_1}, \boldsymbol{h_2}, \boldsymbol{h_3})$ in Figure 2c. This allows the model to infer from the data even in the presence of missing features, in which case we can ignore the subset with missing features. We can also design an aggregation function that computes weighted mean of the representations of subsets since some subsets might be more informative than others:

$$\boldsymbol{h} = \frac{1}{Z} \sum_{k=1}^{K} \eta_k * \boldsymbol{h_k}, \text{ and } Z = \sum_{k=1}^{K} \eta_k, \tag{7}$$

where $K$ is number of subsets, and $\eta_k$ is the weight for $k_{th}$ subset. $\eta$ can be a learnable parameter in semi-supervised, or supervised setting by using an attention mechanism. We can also use 1D convolution in equation (7) by treating representations of subsets as separate channels during training. We left these ideas as future work and used the mean aggregation (i.e. $\eta_k = 1$) throughout our experiments, unless explicitly stated. A comparison of different aggregation methods can be found in Table A3 in the Appendix.

# 3 Experiments

We conducted various experiments on diverse set of tabular datasets including MNIST [27] in tabular format, the cancer genome atlas (TCGA) [42], human gut metagen-omic samples of obesity cohorts (Obesity) [36, 26], UCI adult income (Income) [24], and UCI BlogFeedback (Blog) [4] to demonstrate the effectiveness of the SubTab framework. We compare our method to autoencoder baseline with and without dropout, other self-supervised methods such as VIME-self [46], Denoising Autoencoder (DAE) [43], and Context Encoder (CAE) [39] as well as fully-supervised models such as logistic regression, random forest, and XGBoost [6]. For each dataset, once we decided on a particular autoencoder architecture, we used it for all models compared (i.e. VIME-self, DAE, CAE, and our model). We tried both ReLU and leakyReLU as activation functions for all, and both performed equally well. The code for SubTab is provided[1]. The summary of model architectures and hyper-parameters are in Table A1 in the Appendix. We should note that we ran more experiments using; i) Synthetic datasets and ii) OpenML-CC18 datasets [2] in Appendix G and H respectively.

## 3.1 Data

**MNIST:** We flattened 28x28 images, and scaled them by dividing all with 255 as it is done in [46]. We split training set into training and validation sets (90-10% split) when searching for hyper-parameters, and then used all of training set to train the final model. The test set is used only for final evaluation.

**The Cancer Genome Atlas (TCGA):** TCGA is a public cancer genomics dataset characterized over 20,000 primary cancer and matched normal samples that holds information over 38 cohorts. The task is to classify the cancer cohorts from the reverse phase protein array (RPPA) dataset. It includes 6671 samples with 122 features, which we divided to 80-10-10% train-validation-test sets. Once hyper-parameters is found, we trained the models on combined training and validation set.

**Obesity:** The dataset consists of publicly available human gut metagen-omic samples of obesity cohorts [36]. It is derived from whole-genome shotgun metagenomic studies. The dataset consists of 164 obese patients and 89 non-obese controls and has 425 features [26]. We scaled the dataset by using min-max scaling. Since it is a small dataset, we evaluated the model by using 10 randomly drawn training-test (90-10%) splits, for each of which we used 10-fold cross-validation.

**UCI Adult Income:** It is a well-known public dataset extracted from the 1994 Census database [24]. It includes the details such as education level and demographics to predict whether the income of a person exceeds $50K/yr. The data consists of six continuous and eight categorical features. After one-hot encoding of categorical features, there are total of 101 features. The pre-processing steps can be found in Section B.1 of Appendix.

**UCI BlogFeedback:** The data originates from blog posts, and is originally used for regression task of predicting the number of comments in the upcoming 24 hours. Similar to Yoon et al. [46], we turned it into a binary classification task of predicting whether there is a comment for a post or not.There are 280 integer and real valued features, and separate training and test datasets are provided. Further information can be found in Section B.2 of Appendix.

## 3.2 Evaluation

For self-supervised models, once the models are trained, we evaluate them by training a logistic regression model on the latent representations of training set, and testing it on the latent representation of the test set. For SubTab, the joint latent representation is obtained by using the mean aggregation of embeddings of the subsets for both training and test sets. We use the performance on a classification task as a measure of quality of the representation as it is usually done in the self-supervised learning. MNIST has 10, TCGA has 38, and the rest (i.e. Obesity, Income, and Blog) has 2 classes each.

## 3.3 Results

**MNIST:** We used a simple three-layer encoder architecture with dimensions of [512, 256, 128], referred as the base model, in which the last layer is a linear layer. During training of the base model, we used both reconstruction and contrastive losses. Additionally, we trained our model under three

---

[1]https://github.com/AstraZeneca/SubTab

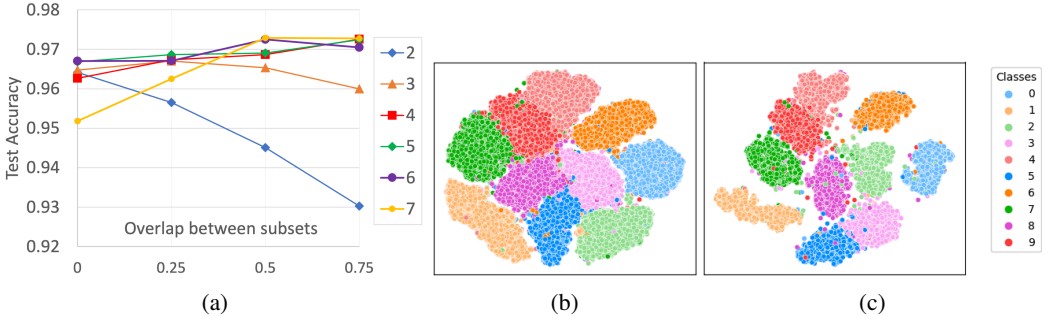

(a)                     (b)                     (c)

Figure 3: a) Test accuracy on MNIST dataset over different number of subsets and varying levels of overlaps. b-c) t-SNE plots for training (b) and test (c) sets of MNIST for the case of using 4 subsets with 75% overlap between neighbouring subsets.

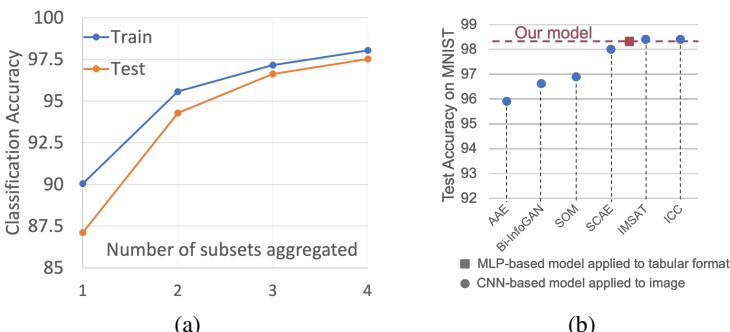

(a)                                  (b)

Figure 4: a) After training the base model (latent dimension=128) on four subsets with 75% overlap, we test its performance using different number of subsets. The performance improves as we start increasing number of subsets involved in prediction. b) Comparing our model to CNN-based SOTA models trained on 28x28 MNIST in image format (please see Section 3.4 for details).

conditions: i) without any noise in the input data, ii) with noise in the input data and iii) same as (ii), but we also added distance loss computed for pairs of projections $\{z_i, z_j, ...\}$.

For SubTab, we trained our base model multiple times without noise at the input. For each training, we used different number of subsets with different levels of overlap between neighbouring subsets (Figure 3a). For small number of subsets (e.g. 2 or 3), the performance monotonically decreases when we increase the overlap between subsets. But, for higher number of subsets, the performance generally improves as we increase the number of shared features between the neighbouring subsets. In general, our results show that $K = 4$ with 75% overlap, and $K = 7$ with 50% overlap perform the best in MNIST dataset, where $K$ refers to the number of subsets. Figure 3 also shows t-SNE plots of training and test sets for K= 4 with 75% overlap, which proves the high quality of clustering, while Table 1 summarizes the classification accuracy of all models on the test set. Our base model without noise outperforms autoencoder baselines and other self-supervised models with the same architecture. We experimented with three noise types for all self-supervised models, and observed that adding swap-noise at the input pushes the performance higher. For SubTab, adding distance loss and increasing the dimensions of the last layer from 128 to 512 help improve the performance even further. Moreover, we conducted three additional experiments (details in Section C.3 of Appendix):

**In the first experiment**, for the optimum case of $K = 4$ with 75% overlap, we trained and tested accuracy of a linear model by using the joint representations obtained from the varying number of subsets. Starting with a single subset of the data, we plot the training and test accuracy of the model (Figure 4a). The linear model is able to achieve 87.5% test accuracy using the representation of a single subset. As we start adding latent representations of remaining subsets, both the training and test accuracy keep increasing, eventually achieving top accuracy when all subsets are used. The evolution of clusters corresponding to Figure 4a can be seen in Figure A7 in Appendix. This experiment indicates that we can achieve a good performance using only small subset of features when we don't have access to data on other features.

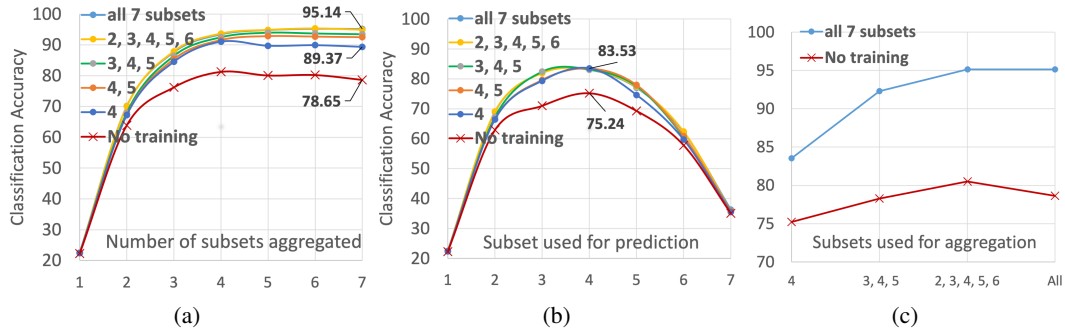

Figure 5: a) The test accuracy using the mean aggregation of the latent representations of subsets, starting with the first subset, and keep adding new subsets sequentially. b) The test accuracy of individual subsets. c) Comparing the test accuracy by aggregating the representations of different set of subsets at test time for two versions of the model; untrained and the one trained on all subsets.

**In the second experiment**, we evaluated SubTab under the condition of missing features during training (Figure 5a). To do so, we first sliced *the unshuffled* features of MNIST to seven subsets with no overlap (the case corresponding to the legend "7" at zero overlap in Figure 3a). Each subset corresponds to four rows in a 28x28 image, starting from top four rows (subset 1) to the bottom ones (subset 7). Then, we trained the base model on five different sets of subsets; $\{4\}, \{4, 5\}, \{3, 4, 5\}, \{2, 3, 4, 5, 6\}$, and $\{1, 2, 3, 4, 5, 6, 7\}$, resulting in five different trained SubTab models. Please note that we selected the sets such that we expand out from the most informative middle regions of the image (i.e. subset 4) to the least informative top and bottom areas.

In order to compare the performance of five models, we followed the following steps for each trained model: 1) We first obtained the embeddings of all seven subsets for both training and test sets; 2) We then trained and evaluated a logistic regression model by using the joint embedding of each of the following seven sets: $\{1\}, \{1, 2\}, \{1, 2, 3\}, ..., \{1, 2, 3, 4, 5, 6, 7\}$ i.e. starting from the first subset, we kept adding new subsets sequentially to increase the information content in the sets. For example, for the set $\{1, 2, 3\}$, we first trained a logistic regression model by using joint embedding of subset 1, 2 and 3 from training set, and evaluated it by using the joint embedding of same subsets from test set. The joint embedding of a set is obtained by using mean aggregation of embeddings of subsets in the set. In addition to five models, we initialized a sixth SubTab model, but kept it untrained and followed the same procedure described before to use it as a baseline. The results are shown in Figure 5a.

In this experiment, we observe that even when the model is trained on a single subset (subset 4, or the blue line in Figure 5a), aggregating the representations of all seven subsets including the subsets not used in training does improve the results. This is because the encoder is able to map samples of different classes to different points in latent space even if it is not trained on them. Since we use the mean aggregation over different views (i.e. subsets) of the same class, we can still make each class in the data distinguishable from the rest in the latent space. We also note that when the model is trained on more and more subsets, its performance keeps improving. As a baseline, we also conducted the same test using untrained model (red line in the plot), and observed similar behaviour in which the test accuracy generally improves as we use more subsets when constructing the joint latent representation. Moreover, we measured the test accuracy of individual subsets to see how informative each subset is (Figure 5b). The result is as expected since we kept the features unshuffled in this experiment, and know that the subsets corresponding to the mid-region of the images (i.e. subsets 3, 4, and 5) should be more informative than the ones corresponding to the top and bottom regions (i.e. subsets 1, and 7). We repeated the same experiment using 28 subsets to get a higher resolution and added the result in Figure A8 in Appendix. From this experiment; i) we see that joint representation improves as we include more subsets (i.e. sub-views) at training and/or test time, ii) we can identify the informative subsets of features using SubTab framework.

**In the third experiment**, we evaluated SubTab on handling missing features at test time. Specifically, we used the model trained on all subsets, and compared it to the untrained model (i.e. our baseline). For each model, we obtained the joint embedding for training set by using mean aggregation over embeddings of all seven subsets, and then trained a linear model. The test accuracy of the linear model is measured by using; i) only subset 4, ii) aggregate of the most informative subsets $\{3,4,5\}$,

Table 1: Accuracy scores for all models for various datasets. The abbreviations in the table; NC: Neighbour columns used, RF: Random features used, G: Gaussian noise used, S: Swap noise used.

| Type | Models | MNIST | Income | Blog | Obesity | TCGA |
|---|---|---|---|---|---|---|
| **Supervised baseline** | Logistic Regression | 92.60±0.03 | 84.68±0.05 | 84.15±0.12 | 62.35±4.02 | 36.98± 1.25 |
| | Random Forest | 96.96±0.06 | 84.62±0.07 | 83.61±0.15 | 67.45±2.23 | 61.62± 1.02 |
| | XGBoost | 98.02±0.086 | 86.11±0.20 | 84.29±0.23 | 64.05±4.52 | 72.61±1.31 |
| **Autoencoder baseline** | AE | 92.77±0.32 | 84.67±0.07 | 84.06±0.24 | 61.96±3.28 | 55.16±0.75 |
| | AE w/ Dropout (p=0.2) | 94.31±0.28 | 85.00±0.10 | 84.18±0.20 | 62.74±4.38 | 56.87±2.26 |
| **Self-supervised** | DAE (RF) | 96.30±0.14 (S) | 84.37±0.36 (G) | 84.12±0.29 (G) | 56.43±5.79 (G) | 54.31±1.39 (G) |
| | CAE (NC) | 96.39±0.20 (S) | 84.24±0.18 (G) | 84.3±0.31 (G) | 62.26±5.01 (G) | 54.20±1.17 (G) |
| | VIME-self | 95.23±0.17 (S) | 84.43±0.08 (G) | 84.11±0.27 (G) | 66.45±4.54 (G) | 55.11±1.37 (G) |
| | **SubTab with:** | | | | | |
| | Base model (No noise) | 97.26±0.2 | 85.31±0.08 | 84.29±0.26 | 68.01±3.07 | 57.02±1.50 |
| | +Noise | 97.47±0.18 (S) | 85.34±0.07 (G) | 84.47±0.15 (G) | **71.13±4.08 (G)** | **58.25±1.36 (G)** |
| | +Distance loss | 97.52±0.14 (S) | **85.35±0.06 (G)** | **84.64±0.19 (G)** | 69.25±4.19 (G) | 58.15±1.56 (G) |
| | +LatentDim=512 | **97.86±0.07 (S)** | - | - | - | - |

iii) aggregate of {2,3,4,5,6} excluding the least informative subsets, and iv) all seven subsets of the test set (Figure 5c).

The results indicate that SubTab can accommodate missing features at test time, and can still perform well. This might also indicate that working with subsets can give us a way to deal with uncertainty better when there are missing features at test time. As the model collects more information in the form of more features, its prediction improves (see Figure 5c). We can also train the model when there are missing subsets during training, and it still performs well (e.g. see legend "4", corresponding to the model trained only on subset 4, in Figure 5a). Our experiments simulate a practical scenario. For example, in healthcare, we might not have access to some features in one hospital while we might have them in another. So, our method would be beneficial in this type of cases.

Overall, we can make the following observations from our experiments: i) the less informative subsets can add value to the overall representation, or at least does not harm the performance (see the aggregate over {3,4,5} versus "All" in Figure 5c), ii) untrained model can be used to analyze which subsets can be potentially more informative, iii) once a model is trained on a subset, the performance of the individual subset does not change whether it is trained together with other subsets or not (for example, compare the performance of subset 3, 4, and 5 across all models in Figure 5b), iv) general idea behind our framework works even for untrained model, and v) we may not need to impute data in our framework since we can simply ignore them as missing subsets, which is good since imputation generally distorts data, and the results.

**TCGA:** We used an encoder architecture with three layers [1024, 784, 784], where the third layer is linear. For VIME-self, DAE, CAE, and our model, we experimented with three noise types (Gaussian, swap, and zero-out noise) at the different % levels of masking ratio $p$. We observed that $p = [0.15, 0.3]$ range worked well for all models. For Gaussian noise, we used a distribution with zero mean, and different levels of standard deviation ($\sigma$). Among all three noise types, Gaussian noise with $\sigma = 0.1$ worked the best for all models. Please note that VIME-self uses swap-noise in its original implementation, but swap-noise does not work well on this dataset. For SubTab, similar to MNIST, we used four subsets with 75% overlap. SubTab performs better than other self-supervised models with a significant margin and almost doubles the performance of logistic regression model trained on raw data as shown in Table 1.

**Obesity:** We used a two-layer encoder with [1024, 1024] dimensions. Second layer is a linear layer. Gaussian noise $\mathcal{N}(0, 0.3)$ and masking ratio $p = 0.2$ works well across all models. Six subsets ($K = 6$) with 0% overlap performed the best for the SubTab. We note that this dataset has 164 obese patients out of 253 total patients. So, the baseline accuracy is $164/253 = 64.82\%$. Based on this fact, we can say that all models, except ours, did not perform well on this dataset. Our model with added Gaussian noise results in accuracy of $71.13 \pm 4.08\%$, which is well above all models, including supervised ones. It means that our model was able to learn useful representation from the data. We should also note that the performance of our model is much better than what Oh and Zhang [36] reported ($66 \pm 3.2\%$) even though they trained a DAE on the same data, and reported their results using a random forest, a non-linear model, on the learned representations rather than a linear model.

**UCI Adult Income & BlogFeedback:** For these two datasets, we used the same architecture as in Obesity. For Income dataset, the best performance is obtained using 5 subsets with 25% overlap whereas we used 7 subsets with 75% overlap for Blog dataset. For the base model, we only used

reconstruction loss. Adding Gaussian noise to the input and distance loss to the objective improves the performance for both datasets. SubTab outperforms other self-supervised models in both datasets.

The choice of hyper-parameters and other details for all experiments can be found in Table A1 in Section C.1 of Appendix.

### 3.4 Ablation study

We conducted a comprehensive ablation study using MNIST. Table 2 summarizes our experiments. The first thing to note is that the performance of the our base model is already good with only reconstruction loss. Hence, we can argue that the reconstruction of original feature space from a subset of features is a very effective way of learning representation. By adding noise to the input data, we can improve the performance. In the case of MNIST, swap-noise is very effective. Also, by adding additional losses such as contrastive, and distance losses as well as increasing the dimension of representation layer from 128 to 512, we can further improve the results. Moreover, we shuffled the features of MNIST to make sure that we don't have any gains from unintentional spatial

Table 2: Ablation study using MNIST with 4 subsets with 75% overlap. Abbreviations are; **RL:** Reconstruction Loss, **CL:** Contrastive Loss, **DL:** Distance Loss, **SF:** Shuffled Features, **LD:** Latent Dim, **Agg:** Aggregating embeddings.

| RL | CL | Noise | DL | SF | LD | Agg | Test Accuracy |
|----|----|-------|----|----|----|-----|---------------|
| + | - | - | - | - | 128 | + | 97.13 |
| - | + | - | - | - | 128 | + | 97.11 |
| + | + | - | - | - | 128 | + | 97.26 |
| + | + | Zero-out | - | - | 128 | + | 97.25 |
| + | + | Gaussian | - | - | 128 | + | 97.25 |
| + | + | Swap | - | - | 128 | + | 97.47 |
| + | + | Swap | + | - | 128 | + | 97.52 |
| + | + | Swap | + | + | 128 | + | 97.2 |
| + | + | Swap | + | - | 512 | - | 95.92 |
| + | + | Swap | + | - | 512 | + | **97.86** |

correlations between neighboring features. We kept all parameters and random seeds same for the comparison. As shown in the table, our model's performance does not change much. We also tried concatenating latent variables of subsets rather than aggregating them when testing the performance. Comparing last two rows in the table, the aggregation is shown to work much better. Please note that we compared different aggregation functions in Appendix F.4, showing that mean aggregation worked the best.

Finally, we compared the performance of SubTab on shallow and deep architecture choices. We trained and tested very shallow architectures for SubTab (referred as shallow SubTab), and compared them to relatively deeper SubTab models used in Table 1 (referred as deep SubTab). We used one-layer encoder and decoder with 784 dimension each for MNIST while using 1024 dimension for other datasets. Shallow SubTab is trained and evaluated under the same conditions as the deeper ones. As shown in Table 3, shallow SubTab significantly improves results in MNIST and TCGA, placing our model performance on par with CNN-based SOTA models [20, 19, 25, 22, 32] as shown in Figure 4b. Obesity is the only dataset which exploits the deeper architecture.

## 4 Related works

We refer the reader to the introduction section that lists some of the recent noticeable works in self-supervised learning. Since our work focuses on tabular data, we will review some of the recent work done in tabular data in self-supervised framework. The most recent work is mostly based on solving a pretext task. For example, Yoon et al. [46] uses a de-noising autoencoder with a classifier attached to its representation layer. A random binary mask is generated to mask and overwrite a portion of entries in the tabular data, and the corrupted data is given as input to the encoder. The classifier is used to predict the mask while decoder is used to re-construct the uncorrupted original input similar to de-noising autoencoder [43]. Although the proposed method is shown to work well in the experiments, there are couple drawbacks to this approach. Firstly, this approach might not work well in very high-dimensional, small and noisy data sets since the model might easily become over-parameterized and be prone to overfitting to the data. Secondly, training a classifier in this

Table 3: Comparing shallow and deep SubTab architectures.

| Model | MNIST | Income | Blog | Obesity | TCGA |
|-------|-------|--------|------|---------|------|
| Deep SubTab | 97.86±0.07 | 85.35±0.06 | 84.64±0.19 | **71.13±4.08** | 58.25± 1.36 |
| Shallow SubTab | **98.31±0.06** | 85.34±0.03 | 84.64±0.09 | 66.88±5.35 | **61.41±1.11** |

setting can be challenging since it needs to predict very high dimensional, sparse, and imbalanced binary mask, similar to the problems observed when training a model on imbalanced, binary dataset. In a similar way, TabNet [1] and TaBERT [45] also tries to recover original data from corrupted one.

## 5 Conclusion

In this work, we show that a simple MLP-based autoencoder trained on MNIST in tabular format can perform on par with the CNN-based SOTA models trained on MNIST images in unsupervised/self-supervised framework. SubTab achieves SOTA in MNIST dataset in tabular setting. We also tested our approach on other commonly used tabular datasets, and proved its benefits. In SubTab, the main performance gain comes from two parts of the model: i) reconstruction of all features from the subset of features, and ii) learning the joint representation by aggregating the embeddings of the subsets.

Using subsets of features may obviate the need for data imputation during training, and allows inference using subsets of features at test time. It might open the door to distributed training of high-dimensional data since the models can be trained on different subsets of features at the same time. We can also potentially take advantage of different datasets with common features by assigning those features to same subsets (i.e. transfer learning). We should note that the subsets shared the same autoencoder in our experiments although we could use separate autoencoders for different subsets if some of the features are drastically different than the rest.

SubTab is computationally scalable when we use only reconstruction loss during training. However, using contrastive, and/or distance losses requires the combinations of projections, which makes the computational complexity quadratic during training and limits the number of subsets we can use to divide the data. In this case, computational complexity is still linear at test time since we need to compute only the aggregate of the representations of the subsets. Also, when we divide the features into subsets, we keep the location of features in each subset same throughout training and test time since neural networks are not permutation invariant. As a possible solution, we can extend our work to permutation invariant architectures by treating collection of features as a set. We also showed that SubTab framework can be used to discover most informative subsets of features with limited resolution. A hierarchical version of SubTab might be used for identifying individual important features, but we leave it as a future work.

Finally, although the primary focus of this work is tabular data setting, SubTab can be extended to other domains such as images, audio, text and so on. We leave the extensions and applications of SubTab as a future work.

## 6 Broader Impact

Tabular data is a commonly used format in healthcare, finance, law and many other fields. Despite its broad usage, the most of the research in deep learning, especially with regards to unsupervised representation learning, has been on other data types such as images, text and audio. Our paper tries to close this gap by introducing a new framework to learn good representations from tabular data in unsupervised/self-supervised setting. The progress in this line of research will open doors to widespread applications of tabular data in other areas such as transfer learning, distributed learning, and multi-view learning, in which we can combine knowledge such as demographics and genomics from tabular data with those in images, text and audio. However, we should be aware of the shortcomings of such data integration in terms of biases and privacy issues that it might introduce.

## 7 Acknowledgements

We thank the anonymous reviewers for their helpful and constructive feedback on the paper. We would also like to thank the entire Respiratory and Immunology AI team and are grateful for general support from other organizations within AstraZeneca.

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
