# A Algorithm

---

**Algorithm 1:** Main learning algorithm

---

**input:** batch size N, constants ($\tau$, $n_s$, is_noise, noise type ), structure of encoder (e), decoder (d);
initialization;

**for** *sampled minibatch* $\{X\}$ **do**
   Divide minibatch $X$ to $K$ subsets $\{x_1, x_2, x_3, ..., x_K\}$ ;
   **if** *Add noise* **then**
      Add noise to each subset in $\{x_1, x_2, x_3, ..., x_K\}$. ;
   **end**
   **for** *Each subset $x_s$ in* $\{x_1, x_2, x_3, ..., x_K\}$ **do**
      `# Forward pass on encoder`
      $h_s = e(x_s)$ ;
      `# Forward pass on projection`
      $z_s = g(h_s)$ ;
      `# Forward pass on decoder`
      $\tilde{X}_s = d(h_s)$ ;
      `# Collect `$h_s, z_s, \tilde{X}_s$
   **end**
   `# We have collected `$\{h_1, h_2, h_3, ...\}$`, `$\{z_1, z_2, z_3, ...\}$` and `$\{\tilde{X}_1, \tilde{X}_2, \tilde{X}_3, ...\}$
   `# Compute reconstruction loss`
   $Loss = \frac{1}{K} \sum_{k=1}^{K} \left( \frac{1}{N} \sum_{i=1}^{N} \left( X^{(i)} - \tilde{X}_k^{(i)} \right)^2 \right)$, where $k \equiv k^{th} subset, i \equiv i^{th} sample$;
   **if** *Apply contrastive or distance loss* **then**
      `# Initialize contrastive and distance losses`
      $\mathcal{L}_c, \mathcal{L}_d = 0, 0$
      `# Generate all combinations of pairs of latents`
      $\{\{z_1, z_2\}, \{z_1, z_3\}, ...\}$ ;
      `# Compute contrastive and distance losses for all pairs`
      **for** *each $j^{th}$ pair $\{z_a, z_b\} \in S = \{\{z_1, z_2\}, \{z_1, z_3\}, ...\}$* **do**
         **if** *Apply contrastive loss* **then**
            # Compute symmetric constrastive loss of pairs $\{z_a, z_b\}$ and update $\mathcal{L}_c$ ;
            $\mathcal{L}_c = \mathcal{L}_c + \frac{1}{2}[l(z_a, z_b) + l(z_b, z_a)]$, where $l(.,.)$ refers to contrastive loss ;
         **end**
         **if** *Apply distance loss* **then**
            # Compute distance loss for each pair $\{z_a, z_b\}$ and update $\mathcal{L}_d$ ;
            $\mathcal{L}_d = \mathcal{L}_d + \frac{1}{N} \sum_{i=1}^{N} \left( z_a^{(i)} - \tilde{z}_b^{(i)} \right)^2$, where i is $i^{th}$ sample in a subset ;
         **end**
      **end**
   **end**
   `# Compute average contrastive & distance losses and update total loss`
   $Loss = Loss + \mathcal{L}_c/J + \mathcal{L}_d/J$, where J is total number of pairs ;
   `# Update network parameters`
**end**
**return** encoder ;

---

# B Data

## B.1 Adult Income Dataset

**Train-Validation-Test Split:** Training and test sets are provided separately [24]. We split the training set into training and validation sets using 80-20% split to search for hyper-parameters. Once hyper-parameters was fixed, we trained the model on the whole training set.

**Features:** The dataset has 14 attributes consisting of 8 categorical and 6 continuous features. We dropped the rows with missing values, and encoded categorical features using one-hot encoding. Features are normalized by subtracting the mean and dividing by the standard deviation, both of which are computed using training set.

**Class imbalance:** It is an imbalanced dataset, with only 25% of the samples being positive.

### B.2 BlogFeedback Dataset

**Train-Validation-Test Split:** The original dataset includes one training set, and 60 small test sets. We combined all the test sets into one test set. We split training set to training and validation using 80-20% split to search for hyper-parameters. We trained the final model using all of the training set.

**Features:** It includes 281 variables consisting of 280 features and 1 target variable indicating the number of comments a blog post received in the next 24 hours relative to the basetime. We converted the target (the last column in the dataset) to a binary variable, in which 0/1 indicates whether the blog post received any comments. We used min-max scaling to normalize the features.

**Class imbalance:** $\sim 36\%$ of the samples are positive in training set while it is $\sim 30\%$ in the test set.

### B.3 Data License

**MNIST** is made available under the terms of the Creative Commons Attribution-Share Alike 3.0 license. **Obesity** is available under MIT license while **Aduld Income** and **BlogFeedback** are under Open Data Commons Public Domain Dedication and License (PDDL).

## C Details of the experiments in the main paper

### C.1 Model architectures and hyper-parameters

Table A1: Architectures & hyper-parameters for the results in Table 1. Abbreviations are; **MNIST***: MNIST with smaller latent dimension, **CL**: Contrastive Loss, **DL**: Distance Loss, **MR**: Mask ratio.

| Dataset | Encoder | Decoder | Projection | DL | CL | Subsets / Overlap | MR | Noise | Batch/Epoch |
|---------|---------|---------|------------|-----|-----|-------------------|-----|-------|-------------|
| **MNIST*** | [512, 256, 128] | [128, 256, 512] | [128] | Yes | Yes, $\tau = 0.1$ | 4 / 75% | 0.15 | Swap | 32, 15 |
| **MNIST** | [512, 256, 512] | [512, 256, 512] | [512] | Yes | Yes, $\tau = 0.1$ | 4 / 75% | 0.15 | Swap | 32, 15 |
| **TCGA** | [1024, 784, 784] | [784, 784, 1024] | [784] | No | Yes, $\tau = 0.1$ | 4 / 75% | 0.2 | Gaussian | 512, 40 |
| **Obesity** | [1024, 1024] | [1024, 1024] | No | No | No | 6 / 0% | 0.2 | Gaussian | 32, 100 |
| **Income** | [1024, 1024] | [1024, 1024] | No | Yes | No | 5 / 25% | 0.2 | Gaussian | 256, 20 |
| **Blog** | [1024, 1024] | [1024, 1024] | No | Yes | No | 7 / 75% | 0.2 | Gaussian | 256, 20 |

Table A2: Architectures & hyper-parameters for the results of Shallow SubTab in Table 3.

| Dataset | Encoder | Decoder | Projection | DL | CL | Subsets / Overlap | MR | Noise | Batch/Epoch |
|---------|---------|---------|------------|-----|-----|-------------------|-----|-------|-------------|
| **MNIST*** | [784] | [784] | [784] | Yes | Yes, $\tau = 0.1$ | 4 / 75% | 0.15 | Swap | 32, 15 |
| **MNIST** | [1024] | [1024] | [1024] | Yes | Yes, $\tau = 0.1$ | 4 / 75% | 0.15 | Swap | 32, 15 |
| **TCGA** | [1024] | [1024] | [1024] | No | Yes, $\tau = 0.1$ | 4 / 75% | 0.2 | Gaussian | 512, 40 |
| **Obesity** | [1024] | [1024] | No | No | No | 6 / 0% | 0.2 | Gaussian | 32, 100 |
| **Income** | [1024] | [1024] | No | Yes | No | 5 / 25% | 0.2 | Gaussian | 256, 20 |
| **Blog** | [1024] | [1024] | No | Yes | No | 7 / 75% | 0.2 | Gaussian | 256, 20 |

**Few other notes:**

- LeakyReLU is used as activation function for all networks.

- Last layers of Encoder, Decoder and Projection shown in Table A1 are all linear.

- Reconstruction loss is used for all experiments by default, except for one case, in which we used only contrastive loss in the ablation study on MNIST to compare it against reconstruction loss (See second row in Table 2).

- Learning rate of 0.001 is used for all experiments since it usually performed the best. Based on that, we optimized the batch size and total number of epochs.

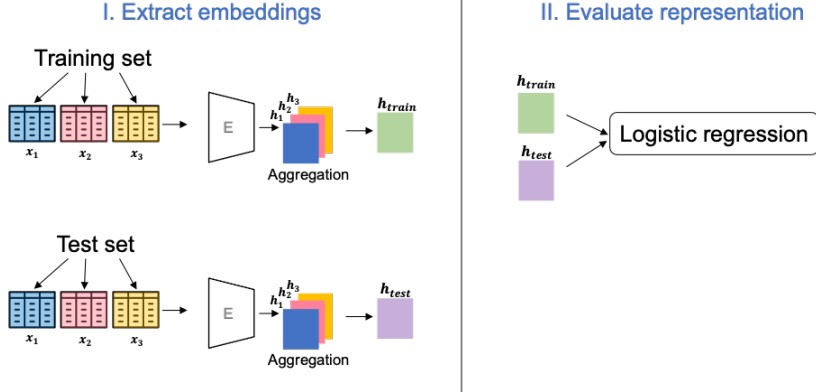

Figure A1: Once the SubTab model is trained, the joint embeddings for both training and test sets are obtained, and the quality of the representation is evaluated by using a linear classifier.

## C.2 Evaluation

For all autoencoder baselines and self-supervised models, we evaluate the quality of the representation by training and evaluating a linear classifier on the embeddings of training and test set respectively. For SubTab, we use the joint embeddings as shown in Figure A1. The models are trained and tested 10 times with different random seeds to compute mean $\pm$ stdev. For the linear model, $l_2$ regularization parameter is selected from a range of 10 logarithmically spaced values in $[10^{-3}, 10^6]$.

## C.3 Supporting visuals for experiments

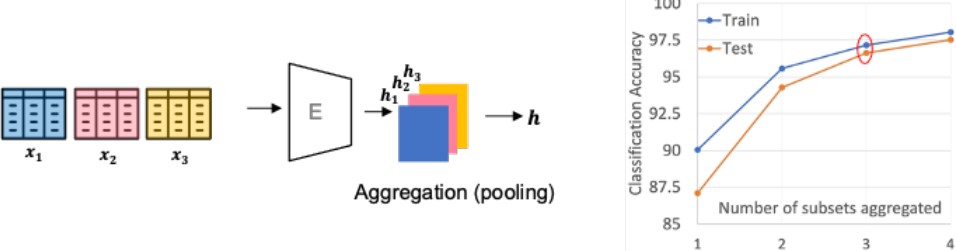

Figure A2: First experiment; measuring the information content of the joint embedding. The example in this figure shows the case for the joint embedding of three subsets.

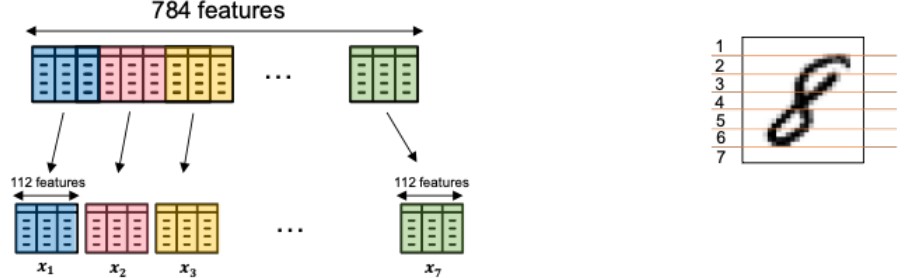

Figure A3: Setup for second and third experiments; MNIST is divided into seven subsets.

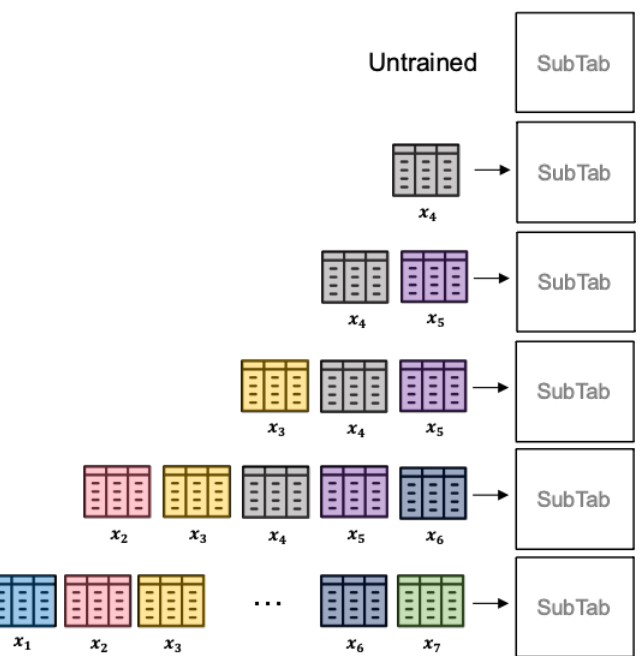

Figure A4: Setup for second experiment, in which five models are trained by using different combinations of seven subsets, and a sixth model that is kept untrained.

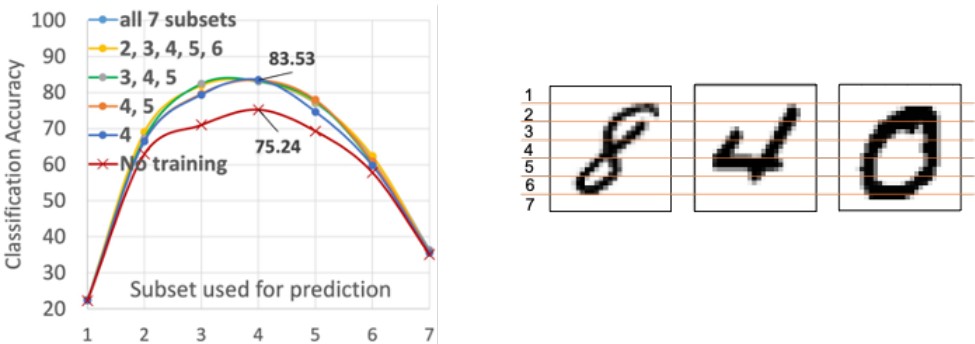

Figure A5: Second experiment; measuring the information content of individual subsets. For each subset, we obtained its embedding from each of the six models for both training and test sets, and evaluated the information content by using a linear classifier.

## C.4 Implementation and resources

We implemented our work using PyTorch [38]. AdamW optimizer [29] with $betas = (0.9, 0.999)$ and $eps = 1e - 07$ is used for all of our experiments. We used a compute cluster consisting of Tesla K80 GPUs throughout this work. SubTab code implemented for MNIST can be found at: https://github.com/AstraZeneca/SubTab.

# D Insights and Comments

## D.1 Insights

- The best performing models are the ones with an over-complete first hidden layer representation. This is also observed in denoising autoencoders [43].
- A simple encoder architecture of [1024, 1024] works well for the most tabular datasets. Note that the second layer is just a linear layer. We can also use a one-layer encoder with 1024 dimension (i.e. removing linear layer).

- It is previously observed that contrastive loss is not stable for small batch sizes [7]. However, we observed that using reconstruction loss together with contrastive loss makes contrastive loss more stable for small batch sizes.

## D.2 Comments

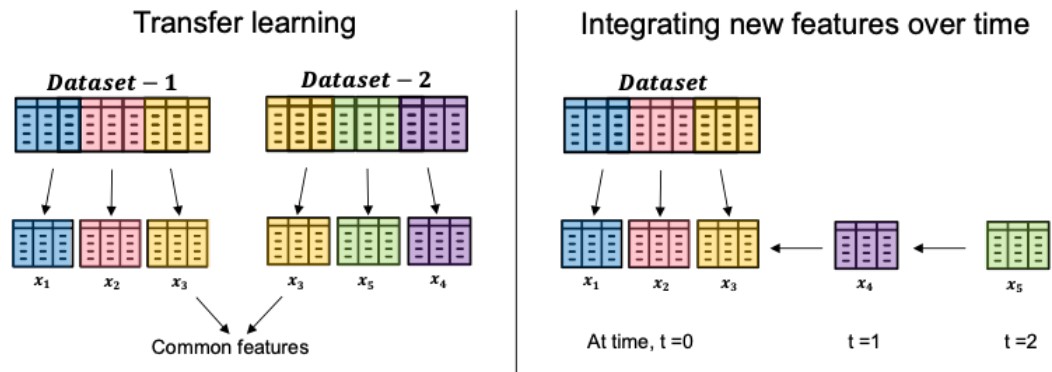

Figure A6: Two of the possible applications of SubTab

**Applications:** Since SubTab works with subsets of features, it can be used in applications such as transfer learning (by using common features across different datasets), few-shot generalisation, domain adaptation, multi-task learning (some subsets might be more useful for one task, and some other for some other tasks), continual learning (adding new features to the dataset to improve performance) and so on in the context of tabular dataset (see Figure A6). Moreover, when there is a distributional shift in certain features, the SubTab can accommodate such situations by relying on the subsets with features that have not changed.

**Other modalities:** In addition to tabular data, SubTab can be extended to other modalities such as images, text, time series, audio and so on by using random subspace of the data.

**Additional limitations:** Dividing tabular data to smaller chunks will result in representation collapse, meaning that the representations of subsets from very different samples might start to have similar representations. However, this is a very low risk since tabular data usually consists of heterogenous features with different statistical properties. Also, using two knobs (the percentage of overlapping features, and number of subsets) further reduces such a risk.

# E  Different configuration of SubTab

## E.1  SimCLR

Removing the decoder, and training the encoder only with contrastive loss would result in the scheme similar to SimCLR. In this case, we can choose to:

- Use two copies of the tabular data (i.e. we are not dividing the features to subsets), and add random noise to each to train the encoder in contrastive learning setting.
- Or use subsets of the tabular data as usual, and train the encoder with constrastive loss. This choice is already shown in the ablation study listed in Table 2.

## E.2  Other choices

We can also choose to use separate encoders for different subsets of the data.

# F  Additional results for the experiments listed in the main paper

## F.1  MNIST

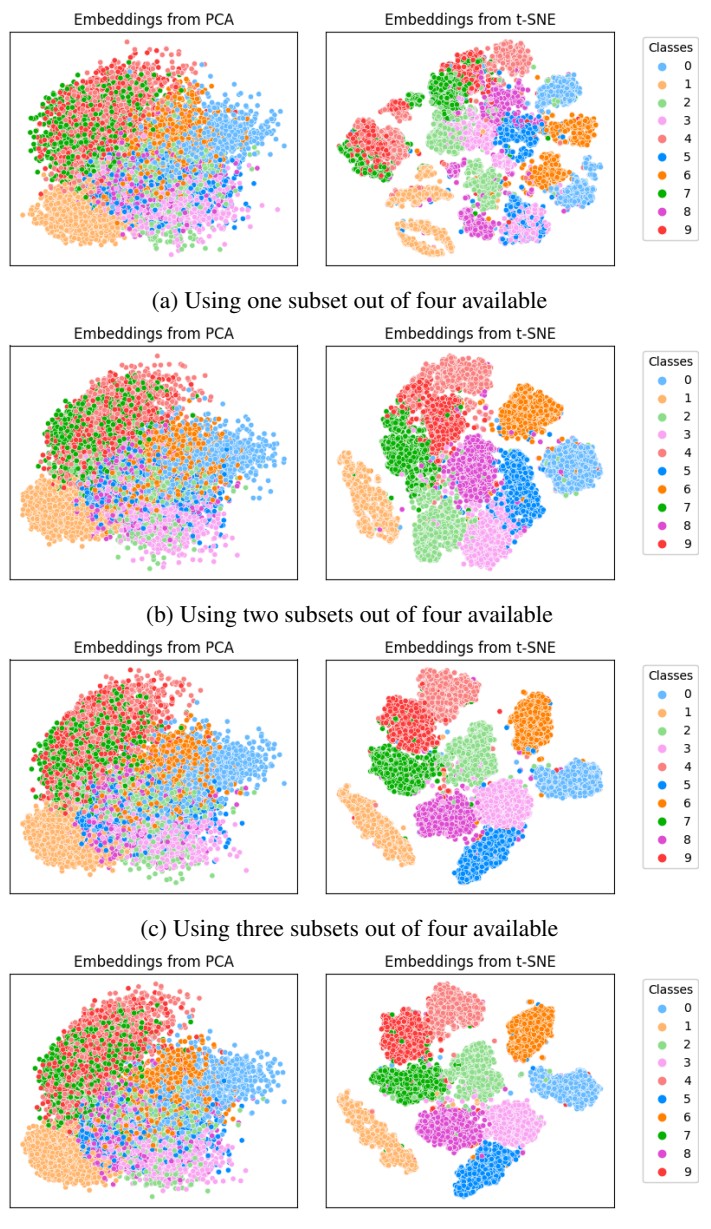

(a) Using one subset out of four available

(b) Using two subsets out of four available

(c) Using three subsets out of four available

(d) Using all four subsets

Figure A7: PCA and t-SNE clustering of representation for the model trained on four subsets with 75% overlap between subsets. Starting from one subset, we keep adding more subsets to get a better representation on the test set: a) One subset, b) Two subsets, c) Three subsets, d) Four (all) subsets.

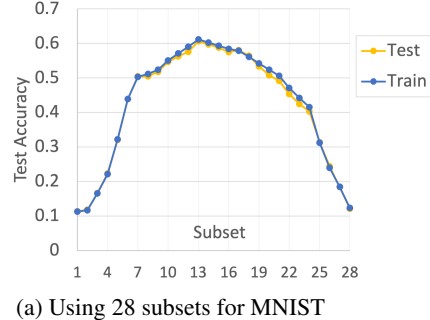

(a) Using 28 subsets for MNIST

Figure A8: a) Dividing MNIST to 28 subsets, each of which corresponds to one row in a 28x28 image. The test accuracy of a subset can be used as a measure of information content in that subset i.e. particular row in the image.

## F.2 Varying the number of subsets and the percentage of overlap for other datasets

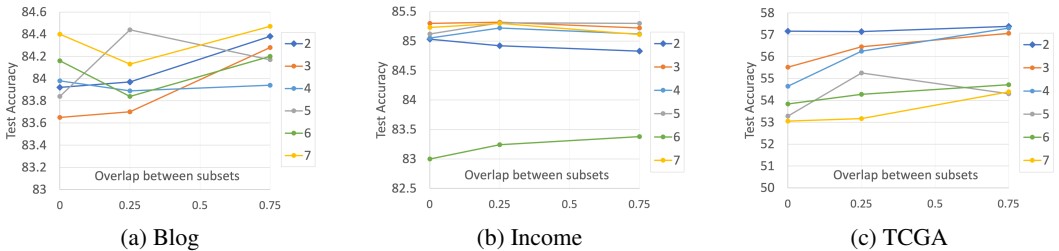

(a) Blog        (b) Income        (c) TCGA

Figure A9: Test accuracy on (a) Blog, (b) Income, and (c) TCGA datasets over different number of subsets and varying levels of overlaps for the base models of each.

## F.3 Sensitivity analysis for masking ratio (p) and initialization

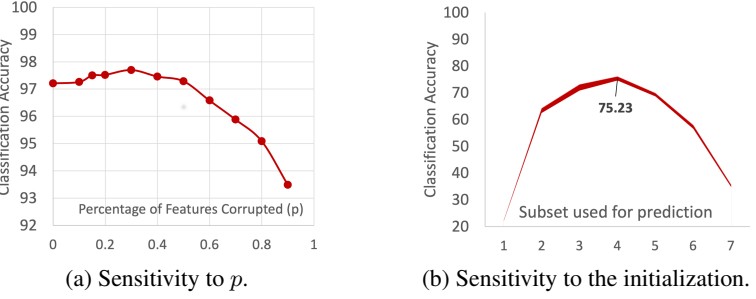

(a) Sensitivity to $p$.        (b) Sensitivity to the initialization.

Figure A10: a) The test accuracy for MNIST over different levels of the masking ratio, $p$. b) Measuring the sensitivity of the model to the initialization by initializing an untrained model with different random seeds and using it to measure the importance of each of the seven subsets for MNIST. This is a repeat of the experiment in the Figure 5.

The sensitivity analysis for the two most important hyper-parameters, i.e. the number of subsets and the overlaps between different subsets, is already shown in Figures 3 and A9. In this section, we show two more sensitivity analysis, one on masking ratio, and one on the initialization of the model.

**Masking ratio** In Figure A10a, we show the sensitivity to the percentages of features corrupted ($p$, or masking ratio) in each subset in our method. For this, we plot the test accuracy for MNIST over different levels of the masking ratio. A good range for $p$ is usually [0.1-0.3], and the model performance is usually robust to the different values of $p$ as shown.

**Sensitivity to initialization** In Figure 5, we showed that we can use untrained model to discover informative subsets of the data. Figure A10b shows how sensitive this analysis is to the initialization

Table A3: Testing classification accuracy of the linear model for the test set of MNIST when using different aggregation functions for the latent representations of subsets.

| Aggregation method | Test Accuracy (%) |
|:---:|:---:|
| **Mean** | **97.86** |
| **Sum** | 97.77 |
| **Max** | 97.79 |
| **Min** | 97.74 |
| **Concatenation** | 95.92 |

Table A4: Summary of three synthetic datasets. Please note that the redundant features are generated using the linear combination of informative ones.

| Dataset | Total features | Informative features | Redundant features | Uninformative features |
|:---|:---:|:---:|:---:|:---:|
| **Dataset-1** | 1000 | 12 | 30 | 958 |
| **Dataset-2** | 100 | 60 | 30 | 10 |
| **Dataset-3** | 100 | 4 | 30 | 66 |

of the network. We initialized the model used for MNIST 10 times with different random seeds, and re-ran the same test of discovering informative subsets. The plot shows the mean test accuracy of the linear model evaluated on the embeddings from the untrained Subtab with 95% confidence interval. The variation is very small, and so we can conclude that the model is not sensitive to the initialization.

### F.4   Using different aggregation functions

We experimented with different aggregations functions when aggregating the latent representations of subsets for the case of MNIST. We also tried concatenating the representations for the downstream classification task using linear model. Table A3 summarizes the result from one such experiment for MNIST. As shown in the table, the performance is robust to the different aggregation methods, but it drops when we use concatenation. Please note that we used mean-aggregation throughout our experiments reported in the paper since it generally performs better.

## G   Experiments using synthetic data

Tabular datasets can be very different from one another in terms of the statistics of the features. Some of them might have many redundant, or uninformative features while some other might have more informative features than the average. Thus, to test the SubTab under different scenarios, we ran more experiments using 3 synthetic datasets that we generated using $make\_classification$ module of scikit-learn library [40].

### G.1   Datasets

Each dataset has 10 classes and 10k samples, 10% of which is used as the test set. We generated them such that the clusters are not easily separable to make the problem more difficult. Specifics of the datasets are summarized in Table A4.

### G.2   SubTab set-up

- The SubTab utilized an encoder architecture of [1024, 1024], of which the first hidden layer uses LeakyReLU, and the second one is a linear layer.

- We used 2 subsets with 25% overlap between them. Other parameters are masking ratio $p = 0.2$, Gaussian noise with $\sigma = 0.1$.

- We trained the model using only reconstruction loss and used mean-aggregation when aggregating the latent representations of the subsets at test time.

Please note that this set-up seems to work well for most tabular datasets as it did in other datasets reported in our work.

Table A5: Summary of the results on the test accuracy (%) for three synthetic datasets.

| Dataset | Raw features | SubTab embedding |
|---|---|---|
| **Dataset-1** | 31.2 | 61.9 |
| **Dataset-2** | 83.5 | 90.5 |
| **Dataset-3** | 79.9 | 82.1 |

Table A6: Summary of datasets from OpenML-CC18. First eight datasets are used for the experiments.

| Dataset | Name | Total Features | Number of Samples | Number of Classes |
|---|---|---|---|---|
| 1 | **First Order Theorem Proving [3]** | 51 | 6118 | 6 |
| 2 | **Wall Robot [12, 14]** | 24 | 5456 | 4 |
| 3 | **Gesture Phase Segmentation [12, 31]** | 32 | 9873 | 5 |
| 4 | **Ozone Level 8hr [12, 47]** | 72 | 2534 | 2 |
| 5 | **Electricity [16]** | 8 | 45312 | 2 |
| 6 | **Texture [18]** | 40 | 5500 | 11 |
| 7 | **DNA [12, 35]** | 180 | 3186 | 3 |
| 8 | **Climate [12, 30]** | 20 | 540 | 2 |
| 9 | Diabetes [33] | 8 | 768 | 2 |
| 10 | Blood transfusion service center [12, 44] | 4 | 748 | 2 |
| 11 | Phoneme [2] | 5 | 5404 | 2 |

## G.3 Evaluation

We trained and tested a logistic regression model on the raw features of the data, as well as the embeddings obtained by using mean-aggregation of the representations of subsets from the SubTab that is pre-trained on the training set. Table A5 summarizes the results on the test set. We can see that the SubTab improves the results in all 3 datasets, as much as 100% for the most difficult dataset (Dataset-1). This result gives us a little bit more insight into how the SubTab might improve the results in the datasets with different nature.

## H Experiments using OpenML-CC18 datasets

Ideally, the type of data we want for the task of representation learning would be a high-dimensional, large dataset with multiple-classes. The most tabular datasets do not fit this criteria. Moreover, OpenML-CC18 [2] includes 72 datasets, in which some datasets have a low number of features (<10) and/or a low number of samples (<1000) such as the last three datasets listed in Table A6:

- Dataset 9: Diabetes

- Dataset 10: Blood transfusion service center

- Dataset 11: Phoneme

Thus, we excluded such datasets from consideration, and picked other eight datasets in Table A6 for experiments. Please note that although Electricity dataset has only 8 features, we included it in our analysis since it does have relatively large sample size.

### H.1 Data Pre-processing

We cleaned up the datasets by removing rows with missing data if there is any, and/or by removing the features such as user ID. We used min-max scaling to scale all datasets, and split the data as 70-10-20% training, validation and test set. We trained the final models on 80% training set by combining training and validation set.

### H.2 Models and Evaluation

We trained and compared six models: i) Logistic Regression as our baseline, ii) Autoencoder (AE) iii) Autoencoder (AE) with dropout (p=0.04), iv) VIME-self, v) SubTab, and vi) SubTab with dropout (p=0.04).

Table A7: The results for the eight datasets. Please refer to Table A6 for the name of the datasets.

| Model | Dataset-1 | Dataset-2 | Dataset-3 | Dataset-4 | Dataset-5 | Dataset-6 | Dataset-7 | Dataset-8 |
|---|---|---|---|---|---|---|---|---|
| Logistic Regression | 46.96 | 68.46 | 46.93 | 94.01 | 76.09 | **99.71** | **95.12** | **96.3** |
| Autoencoder (AE) | 50.40±0.83 | 86.83±0.91 | 49.07±0.55 | 94.84±0.34 | 81.32±0.16 | 99.34±0.28 | 93.48±0.97 | 95.01±0.90 |
| AE w/ Dropout (p=0.04) | 50.52±0.71 | 86.87±0.44 | 49.43±1.17 | 94.69±0.14 | 81.54±0.36 | 98.75±0.18 | 91.48±0.43 | 95.04±1.06 |
| VIME-self | 44.99±0.9 | 74.23±1.21 | 46.08±0.37 | 94.28±0.31 | 73.92±1.08 | 95.49±0.88 | 89.97±0.97 | 95.56±0.42 |
| SubTab | 50.8±0.76 | 89.37±0.72 | **50.33±0.86** | 94.74±0.28 | 82.11±0.26 | 99.59±0.22 | 92.62±0.59 | 93.89±1.55 |
| SubTab w/ Dropout (p=0.04) | **51.48±0.77** | **89.81±0.69** | 49.93±0.77 | **94.85±0.31** | **82.31±0.34** | 99.23±0.36 | 91.41±1.03 | 93.33±0.77 |

All neural networks used the same four-layer encoder architecture: [256, 256, 256, 256]. For the networks with dropout, we used the same dropout rate, p=0.04. We trained SubTab by using two subsets with zero overlap and using only reconstruction loss. For all models using neural networks, we trained and evaluated them with 10 different random seeds. Evaluation of these models is done by training a logistic regression model using the embeddings of training set (i.e. 80% of the data), and by testing it using the embeddings of the test set (20% of the data).

## H.3 Results

Based on the results shown in Table A7, we can make following observations:

- If the dataset is suitable for pre-training / representation learning, the SubTab tends to perform better than the other approaches, including VIME-self [46]. This was the case in the datasets [1-5].

- If the dataset is trivial (i.e. logistic regression already gives a very decent performance), we might be better off using simple models such as logistic regression as this was the case in datasets [6, 7, and 8] (i.e. Texture, DNA, and Climate datasets respectively).