# OpenReview forum: "SubTab: Subsetting Features of Tabular Data for Self-Supervised Representation Learning"
_NeurIPS.cc/2021/Conference — NeurIPS 2021 Poster_

### Official Review · Reviewer_b2iX · 2021-07-08

**Rating:** 6
**Confidence:** 4

**Summary:**

The authors propose a novel self-supervised learning for tabular data.
The authors utilize the reconstruction of the entire data using the subsets of the features as the pretext task.
The authors show some consistent performance improvements in comparison to SOTA SSL in tabular representation learning.

**Ethical Concerns:**

There is no ethical concern.

**Limitations And Societal Impact:**

The authors need to illustrate the limitation of the proposed model explicitly.
For instance, the proposed method cannot provide the feature level importance.
There is no negative social impact.

**Main Review:**

1. Hyper-parameters-
 It seems like the proposed method introduces multiple hyper-parameters: alpha, beta, gamma, p, etc.
- How to optimize those hyper-parameters? What is the objective function to optimize them?
- It would be good if the authors can show some sensitive analyses about those hyper-parameters.
- If eta is a learnable parameter, how can we learn the optimal eta? It would be good if the authors can describe that clearer.

2. Neighbor columns
- In image data, neighbor pixels should share the similar patterns.
- On the other hand, in tabular data, there is no reason to share some patterns across neighbor features.
- At that point, what is the difference between neighbor columns and random columns? We can still make some overlapping in random columns case.

3. Ensemble
- How can this work be related to the ensemble works?
- It seems like using weighted sum among the different representations can be interpreted as the ensemble approach for representation learning.
- As the authors said, eta_k = 1 for all k. In that case, it seems like a similar idea with bagging.

4. Discovering the informative features
- It seems like the proposed model only determines which subset of features is more important.
- However, it cannot show a feature level importance.
- It would be good to describe this kind of limitation.
- Also, it would be good if the authors provide some qualitative analyses on this feature subset importance in the experiment section.

5. Etc
- How much performance differences between ReLu and LeakyReLu? Also, can the authors show the performances of VIME-self with LeakyReLu?
- How is the performance of the proposed model with ReLU? (for fair comparison with VIME)

**Time Spent Reviewing:**

4 hours

---

> ### Author Response · Authors · 2021-08-10
> **Response to Reviewer b2iX (Part I)**
>
> We thank the reviewer for the thoughtful comments and finding our method novel. Our response is below:
>
> ---
> >**1.	The reviewer: Hyper-parameters- It seems like the proposed method introduces multiple hyper-parameters: alpha, beta, gamma, p, etc. How to optimize those hyper-parameters? What is the objective function to optimize them?**
>
> **Our response:**
>
> - Many thanks for bringing up this point of confusion. We included hyper-parameters in the formulation just to keep equations in a very general form. But, in our implementation, we never used the parameters defined in objective functions and aggregation (averaging of the latents), namely alpha, beta, gamma, eta in equations 2, 3, 4 and 6 (i.e. they were all equal to 1).
>
> - As it can be seen from the ablation study (Table-2 in the paper), and results in Table-1, the main gains in performance mainly came from:
>
>   - I.	Reconstruction of all features from its subsets
>   - II.	Aggregation of the representations of subsets
>
> - Main hyper-parameters that we used are:
>   1.	Type of noise (3 of them)
>   2.	p , ratio of features corrupted in the subset (we used the range [0.1-0.3] )
>   3.	In the case of Gaussian noise, we used one parameter for variance (i.e. noise level), which kept it at 0.1 for all experiments that involved Gaussian noise.
>   4.	Number of subsets, for which we showed our sensitivity analysis in Figures 3a in the paper, and Figure-7 in the supplementary material.
>   5.	Overlap % between subsets (i.e. number of shared features between subsets), for which we showed the sensitivity in the same figures mentioned above.
>
> - Compared to other methods, our method introduced only two hyper-parameters: “Number of subsets”, and “overlap between subsets”, which were the natural result of our proposal. Since the models compared are all autoencoder-based models, the major difference in performance came from how they augment the data.
>
> ---
> >**2.	 The reviewer: It would be good if the authors can show some sensitive analyses about those hyper-parameters.**
>
> **Our response:**
>
> - We show how performance of the SubTab changes when we sweep the percentage of the overlap between the neighbours as well as number of subsets for various datasets (Figure-3a, and 7)
>
> - For p, ratio of input features corrupted, the values between 0.15-0.3 gave similar performance, but degraded below and above it in general.
>
> - We did run additional sensitivity analysis on the masking ratio ( p ), using MNIST. As shown in the result below, the method is not sensitive to ratio of corrupted features in the range [0.15-0.3], which was within our sweeping range when optimising for different datasets:
>
> | p             	| 0     	| 0.1   	| 0.15 	| 0.2   	| 0.3  	| 0.4   	| 0.5   	| 0.6   	| 0.7   	| 0.8   	| 0.9   	|
> |---------------	|-------	|-------	|------	|-------	|------	|-------	|-------	|-------	|-------	|-------	|-------	|
> | Test Accuracy 	| 97.21 	| 97.26 	| 97.5 	| 97.52 	| 97.7 	| 97.46 	| 97.29 	| 96.58 	| 95.88 	| 95.09 	| 93.49 	|
>
> - Table-2 in our paper compares the model performance with and without noise, as well as with different types of noise. Adding noise contributed to the performance very little in this case.
>
> - Similarly, Table-1 compares the SubTab performance with and without noise for all datasets. For most datasets, the contribution of noise to the performance was =< 1% for SubTab.
>
> - In our work, the most important (hence sensitive) hyper-parameters was number of subsets and overlap between neighbours. Overall, the major gains come from two factors:
>   - Learning representation by reconstructing of all features from subsets
>   - Aggregating the representation of the subsets.
>
>
>
> ---
> >**3.	The reviewer: If eta is a learnable parameter, how can we learn the optimal eta? It would be good if the authors can describe that clearer.**
>
> **Our response:**
>
> - As pointed out in our response to the first question, Eta is not used as a learnable parameter in this work, and we just used 1 for etas as for the most of other hyper-parameters defined in the equations. We will keep “learnable eta” for future work.
>
> - There are few ways of implementing learnable eta in the future. In our paper, we aggregate the representations of the subsets to learn a joint representation during test-time, which is analogue to the aggregation of node representations in Graph Neural Networks [1], or pooling operation in Computer Vision.
>
> - One option is that we can use 1D convolution to compute the aggregate embedding by treating the embeddings of the subsets as separate channels and use the aggregate to compute reconstruction loss during training.
>
> - As a second option, we think that one can use attention mechanism to learn eta’s during training in semi-supervised, or supervised setting, similar to self-attention mechanism used to attend the neighbouring nodes in Graph Networks [2].
>
> ---
> >**4.	The reviewer raise a question about the Neighbour columns, i.e. sharing features between subsets in our paper, raising two points:
> •	In image data, neighbor pixels should share the similar patterns.
> •	On the other hand, in tabular data, there is no reason to share some patterns across neighbor features.**
>
> **Our response:**
>
> - Our method of dividing features to subsets with overlap between them is similar to the feature bagging in random forest. In random forest, it is done for classification task while the SubTab does it for representation learning. This is also analogue to cropping images, and zooming-in Computer Vision.
>
> - Although there might not be any spatial or semantic relation between neighbour features (or shared features), we used it as a knob to increase the correlation between subsets. As it can be seen in our experiments (Figure-3a, Figure-7), where we plot performance across different levels of the overlap, higher correlation between different subsets does tend to improve the results, but it also depends on how we divide the data as well.
>
> ---
> >**5.	The reviewer At that point, what is the difference between neighbour columns and random columns? We can still make some overlapping in random columns case.**
>
> **Our response:**
>
> - Yes, we agree with the statement. As stated earlier, our method is similar to the feature bagging in random forest. One can assign features to different subsets during the data preparation step. This is also equivalent to shuffling location of features first, and then dividing them into subsets with overlapping regions between neighbour subsets.
>
> - We used the term “neighbour” columns to avoid any confusion, and to give the reader an analogue to the data types such as images. But, the key factor is that once the features are assigned to corresponding subsets, the location of features in each subset cannot be changed during the training and at test-time since neural networks are not permutation invariant.
>
> ---
> >**6.	The reviewer raises a question on similarity between averaging of embeddings of subsets to Ensembling, and asks following questions:
> •	How can this work be related to the ensemble works?
> •	It seems like using weighted sum among the different representations can be interpreted as the ensemble approach for representation learning.
> •	As the authors said, eta_k = 1 for all k. In that case, it seems like a similar idea with bagging.**
>
> **Our response:**
>
> - Our method is not an ensembling method, but is rather an analogue to mean-aggregation function used in Graph Neural Networks [1], in which a node’s representation is computed using aggregate of its neighbours. It is also similar to the mean-pooling operation commonly used in Computer Vision.
>
> - The novel ideas behind our paper were:
> 	- i) To generate sub-views from the tabular dataset (i.e. subsets) through feature-bagging,
> 	- ii) The aggregation of the latent representations of the sub-views. For aggregation, we choose to use mean-aggregation (i.e. averaging), but one can choose to use min, max, sum, and other pooling operations.
>
> - We also would like to make one more distinction: We use the aggregation to learn a representation rather than making a prediction. Ensemble learning is done using predictions of multiple models either explicitly (using either multiple different models, or using an ensemble model such as Random Forests), or implicitly (e.g. by using dropout):
>
>   - In ensembling by using multiple models, or using dropout method, the models are trained on the same set of features. And averaging is done at the output. In contrast, each subset in our method has different features, and aggregation is done to get an aggregate representation of the data before using it in a downstream task such as classification.
>
>   -	In the case of Random Forest, it does use feature-bagging, similar to our method of subsetting features, but RF can be considered as training multiple models. The averaging in Random Forest is done again by averaging the predictions of multiple sub-models (trees).
>
> - A more insight into what the aggregation is doing can be seen in Figure-6 in the Supplementary section, in which we show t-SNE plots of clusters of MNIST digits, and we compare the clusters obtained using the embeddings of 1 subset, the aggregate of 2 subsets and so on. As we aggregate more of the subsets, the overall representation is being improved.
>
> - Moreover, we ran an experiment using SubTab on MNIST. Once the SubTab is pre-trained, we evaluate the embeddings of the test set using a linear model that is trained on the embeddings of training set.  We obtain embeddings by using different aggregation methods for combining embeddings of the subset, and below are the results (Please note that we also tried concatenating the latent embeddings of the subsets as shown in the last row):
>
>    | Aggregation method | Test Accuracy |
>    |--------------------  |---------------  |
>    | Mean | **97.87** |
>    | Sum  | 97.77 |
>    | Max  | 97.79 |
>    | Min  | 97.74 |
>    | Concatenation  | 95.92 |
>    |  | |
>
>   We hope that this clarifies the difference.

---

> > ### Author Response · Authors · 2021-08-10
> > **Continuing our response to Reviewer b2iX (Part II)**
> >
> >
> > ---
> > >**7.	The reviewer raises a point about discovering the informative features:
> > •	It seems like the proposed model only determines which subset of features is more important.
> > •	However, it cannot show a feature level importance.
> > •	It would be good to describe this kind of limitation.**
> >
> > **Our response:**
> >
> > - We have not designed SubTab for the purpose of finding important features. But, its side-benefit is that we can discover important regions of the features, which can be very important in very high-dimensional data such as the ones in genomics. However, it is limited as stated since dividing subsets to smaller chunks will result in representation collapse, meaning that the representations of subsets from very different samples might start to have similar representations.
> >
> > - We will update the paper, mentioning its limitation in feature importance.
> > - **Side note:** we believe that one can use our method hierarchically to find important features – perhaps a future work.
> >
> > ---
> > >**8.	The reviewer: Also, it would be good if the authors provide some qualitative analyses on this feature subset importance in the experiment section.**
> >
> > **Our response:**
> >
> > - We did give some insights in the case of MNIST data. In the original form of MNIST images (features unshuffled), the mid-region of the image is the most informative while top and bottom regions would be the least informative sections. We divided the features to 7 subsets, and we measured the informativeness of 7 subsets used in MNIST results (Figure 5b). It is meant to show the informativeness of each individual subset as well as the informativeness of their aggregate representation before and after training the model. When aggregating the latent embeddings of subsets, we tried to show how their predictive power varies as we use different combination of subsets to get the aggregate representation.
> >
> > - Additionally, we will add one more experiment in the Supplementary section, in which we increased number of subsets to 28. So each subset corresponds to a single row in a 28x28 MNIST image. And we measured the informativeness of each subset using their test accuracy. The results show that the most informative subset was #13, corresponding to row 13 in 28x28 image as expected.
> >
> > - We will update the paper to give more insights. If it does not fit in the main paper, we will do it in the supplementary material. Please let us know if there is anything specific you would like to see.
> > ---
> > >**9.	The reviewer:
> > •	How much performance differences between ReLu and LeakyReLu? Also, can the authors show the performances of VIME-self with LeakyReLu?
> > •	How is the performance of the proposed model with ReLU? (for fair comparison with VIME)**
> >
> > **Our response:**
> >
> > - We ran experiments using the same encoder architecture for both methods. For MNIST, we used the encoder with [512, 256, 128], and as shown below, the difference in performance between using ReLU and LeakyReLu is not significant.
> >
> >
> >    |                       	| MNIST          	|
> >    |-----------------------	|----------------	|
> >   | VIME with ReLU        	| 94.67 +/- 0.18 	|
> >   | VIME with LeakyReLU   	| 94.77 +/- 0.12 	|
> >   | SubTab with ReLU      	| **97.32 +/- 0.13** 	|
> >   | SubTab with LeakyReLU 	| **97.52 +/- 0.14** 	|
> >
> >   In our method, the main performance gain came from two factors:
> >   - Reconstruction of all subsets from the subset of features
> >   - Aggregation of the embeddings of the subsets to get the joint representation.
> >
> >
> > ---
> > **References:**
> >
> > [1] Semi-Supervised Classification with Graph Convolutional Networks: https://arxiv.org/pdf/1609.02907.pdf
> >
> > [2] Graph Attention Networks, https://arxiv.org/pdf/1710.10903

---

### Official Review · Reviewer_TMPY · 2021-07-14

**Rating:** 7
**Confidence:** 4

**Summary:**

This paper proposes a self-supervised framework for tabular data. The idea is to split tabular data into multiple subsets of features, embed them into a latent space, and compute a joint representation by averaging the subsets’ latent variables. The approach outperforms existing self-supervised tabular baselines and reaches performances that are on par with CNN models on MNIST.

**Limitations And Societal Impact:**

The authors adequately addressed the limitations and potential negative social impact of their work.

**Main Review:**

Strengths:
- The proposed method is simple, elegant, and conceptually appealing. The approach has many interesting applications such as learning better representations in tabular datasets (e.g. for healthcare) and distributed training.
- The approach outperforms existing self-supervised methods in a number of tabular datasets. It also reaches performances that are on par with CNN models on MNIST.
- The authors analyse the performance of the model across different numbers of subsets, overlap rates, types of noise, and trained/untrained models, with interesting insights and practical advice (e.g. the untrained model can be used to detect informative subsets).

Weaknesses:
- It is mentioned that “all subsets are fed to the same encoder to get their corresponding latent representation” (L65-66) and that “we don’t change the relative order of features in a subset since standard neural network architectures are not permutation invariant” (L80-81). It is unclear how this happens - do the authors mask out features of other subsets (e.g. by setting them to 0)? If that’s the case, on what basis do they claim that SubTab allows to “use smaller models by reducing input dimension, making it less prone to overfitting“ (L53)? Or do they reuse the same input neurons for different input features?
- The claim that SubTab can “do training and inference in the presence of missing features by ignoring corresponding subsets” (L52-53) is not well supported in the paper. Can the authors demonstrate this? While SubTab might work well in some scenarios (e.g. when the variables are missing completely at random; MCAR), in many real-world scenarios the MCAR assumption does not hold. For example, for single-cell RNA-seq, it is well known that dropout patterns are informative of cell-type (e.g. see https://www.nature.com/articles/s41467-020-14976-9) and thus ignoring this information can result in degraded performance.
- How does the proposed approach compare to a plain autoencoder with the same encoder/decoder architectures? How does it compare to self-supervised TabNet? (see https://arxiv.org/abs/1908.07442)

Minor comments/questions:
- “We shuffle the order of subsets in every batch during training to avoid introducing any unintentional bias“ (L79-80). How exactly would the order introduce a bias? The forward pass of SubTab appears to be invariant to subset permutations.
- Equations 4 and 5 might be incorrect and can potentially be simplified. 1) The indices a and b are undefined - don’t the authors loop over the subset indices a and b? (similar for Equation 6), 2) isn’t the overall contrastive loss computed by averaging the individual positive (pairs of subsets within the same sample) and negative (pairs of subsets within different samples) losses? Equation 4 seems to imply that the loss is computed only on contiguous samples within the batch (in which case the order of subsets might indeed result in an unintentional bias).
- How did the authors optimise the hyperparameters of the baselines? To avoid biases in the comparison, all methods should be optimised in the same way.
- Results for some baseline methods are taken from Yoon et al. “Extending the success of self-and semi-supervised learning to tabular domain”. Did the authors ensure that the train/validation/test splits are the same?
- The untrained model achieves better than random performances. How did the authors initialise the weights? How sensitive is the initialisation? It would be helpful to provide the standard error in Figure 5 (and all figures in general).
- Intuitively, it seems that SubTab should work well with highly redundant datasets where several sets of features are informative of the underlying latent variables (e.g. MNIST or TCGA). How does the model perform when this is not the case? Do the claims from L49-L53 still hold?


**Time Spent Reviewing:**

5

---

> ### Author Response · Authors · 2021-08-10
> **Response to Reviewer TMPY (Part I)**
>
> We thank the reviewer for insightful feedback and finding our method simple and elegant.
> Our response is below:
>
> ---
> > **1. The reviewer refers to the lines “all subsets are fed to the same encoder to get their corresponding latent representation” (L65-66) and the lines “we don’t change the relative order of features in a subset since standard neural network architectures are not permutation invariant” (L80-81), and ask whether we re-use the same input neurons for different input features.**
>
> **Our response:**
>
> - Yes, we re-use the input neurons for different input features (i.e. parameter sharing). To simplify, we have 784 features in MNIST. If we had 7 subsets with no overlap, each subset would have 784//7 = 112 features. So, input features of our network would be 112.  All 7 subsets share the parameters of the network.
>
> - In applications in which we have very high dimensional data, such as in genomics, we think that reducing the effective input size might help with some of the issues with over-parameterized models especially if the dataset size is small. If the input dimension is reduced, we can also reduce the rest of the layers of the model.
>
> ---
> > **2. The reviewer refers to our claim that the SubTab can “do training and inference in the presence of missing features by ignoring corresponding subsets” (L52-53) and thinks that it is not well supported in the paper, and asks us whether we can demonstrate it.**
>
> **Our response:**
>
> - We demonstrated this claim in Figure-5b and 5c. For example, the blue line with label “4” in Figure-5b corresponds to a case, in which we trained the model using only subset-4, meaning that subsets [1, 2, 3, 5, 6, 7] were not used during training, and assumed that we did not have access to them. Then, we evaluated the model trained on Subset-4 by using the embeddings of each of the other subsets to test their predictive power even though they were not used during training. As our baseline, we compared it to the embeddings of the subsets obtained from the untrained model.
>
> - If we compare the predictive power of subset-3, by using its embeddings obtained from a model trained on subset-4 and the ones from an untrained model, we observe that the model trained on subset-4 still gives a more informative embeddings about subset-3.
>
>    And we go on to repeat this experiment using 4 other models trained on different combinations of the subsets: [4, 5], [3, 4, 5] and so on.
>
> - Moreover, in Figure-5c, we have a model trained on all the subsets. But we ask what happens if we only have access to subset of features at test time. Thus, we measure the predictive power of joint representation of available subsets e.g. the case when we only had access to subset-4, subsets [3, 4, 5], and so on. We show that , at test-time, even when we have access to a single subset (subset-4), we still benefit from it. And our baseline was the embeddings from the untrained model again to show the difference.
>
> - We understand that our experiment is rather an extreme case, compared to randomly missing features in various samples. But we suggested that one can choose to mask a particular subset of a particular sample if there are missing features in the subset. This is rather a better situation than the one we experimented in the paper since the model would still have access to the subset in other samples.
>
> - Our focus in this experiment was to simulate a real-life scenario. For example, in healthcare, we might not have access to certain features in one hospital while we might have them in another. So, our method would come handy in these cases.
>
>   We hope that this clarifies our claim. If there is a further question, please let us know.
>
> ---
> > **3. The reviewer:   While SubTab might work well in some scenarios (e.g. when the variables are missing completely at random; MCAR), in many real-world scenarios the MCAR assumption does not hold. For example, for single-cell RNA-seq, it is well known that dropout patterns are informative of cell-type (e.g. see https://www.nature.com/articles/s41467-020-14976-9) and thus ignoring this information can result in degraded performance.**
>
> **Our response:**
>
> - Our method is orthogonal to any data pre-processing step, thus our method would work fine with the data that is imputed, noisy, or is with missing features that is binarized as it is done in the cited nature paper. We were only suggesting that one can optionally choose to mask out (i.e. ignore) individual rows with missing features of a subset during training, or inference. If the missingness pattern is informative about the underlying process, our model should be able to take advantage of it.
>
>
> ---
> > **4. The reviewer wonders how our approach compares to a plain autoencoder with the same encoder/decoder architectures? How does it compare to self-supervised TabNet?**
>
> **Our response:**
>
> - **About plain autoencoder:**
> Our work focuses on representation learning through data augmentation. Hence, we considered models that take advantage of various data augmentation schemes. What differentiates this class of models such as De-noising Autoencoder from a plain autoencoder is that they are not constrained to have a bottleneck layer. Since the input data is augmented, there is no risk of the model learning an identity function. However, plain autoencoders have this risk if there is no bottleneck. Please note that the models in our paper usually don’t have a bottleneck, hence making the comparison to plain AE difficult.
> However, for the rebuttal, we ran this experiment for plain AE using the same architecture as SubTab for each dataset, and trained and tested the model with 10 different random seeds:
>
>    | Dataset 	| plain AE       	| SubTab         	|
>    |---------	|----------------	|----------------	|
>    | MNIST   	| 92.77 +/- 0.12 	| **97.86 +/- 0.07** 	|
>    | Income  	| 84.67 +/- 0.07 	| **85.35 +/- 0.06** 	|
>    | Blog    	| 84.06 +/- 0.24 	| **84.64 +/- 0.19** 	|
>    | Obesity 	| 61.96 +/- 3.28 	| **71.13 +/- 4.08** 	|
>    | TCGA    	| 55.16 +/- 0.75 	| **58.25 +/- 1.36** 	|
>    |         	|                	|                	|
>
>
>
>
>
> - **About TabNet:**
>  In this work, our main focus was on autoencoder-based approaches for unsupervised representation learning, namely; Denoising Autoencoder, Context Autoencoder, and VIME (which is a Denoising Autoencoder with additional classifier for mask prediction).
>
>   TabNet itself is a sequential model. Moreover, pre-training via self-supervised learning in TabNet seems to be equivalent to VIME. TabNet tries to predict masked features, same as VIME. VIME authors compared their method to TabNet  in Table 5 of the supplementary material of their NeurIPS paper [1], and showed that TabNet performs similar to VIME. Given the results in their report, we can say that our method should outperform TabNet in MNIST, Blog and Income datasets. So, we would expect that the representation learned in our method should be better than the one learned in self-supervised pretraining of TabNet in general.
>
> ---
> > **5. The reviewer points out to lines “We shuffle the order of subsets in every batch during training to avoid introducing any unintentional bias“ (L79-80), and asks how the order introduces a bias, considering that the forward pass of SubTab appears to be invariant to subset permutations.**
>
> **Our response:**
> - We agree with the reviewer,  it should have no effect indeed. We will update the paper.
>
>
> ---
> > **6. The reviewer: Equations 4 and 5 might be incorrect and can potentially be simplified. 1) The indices a and b are undefined - don’t the authors loop over the subset indices a and b? (similar for Equation 6), 2) isn’t the overall contrastive loss computed by averaging the individual positive (pairs of subsets within the same sample) and negative (pairs of subsets within different samples) losses? Equation 4 seems to imply that the loss is computed only on contiguous samples within the batch (in which case the order of subsets might indeed result in an unintentional bias).**
>
> **Our response:**
> - We agree with the reviewer. (a,b) is meant to refer to a pair of subsets, and we have J number of them. Also, the reviewer’s initial understanding is correct, and the order of subsets has no effect. We will make the changes to make the formulation clearer.
>
>
> ---
> > **7. The reviewer: How did the authors optimise the hyperparameters of the baselines? To avoid biases in the comparison, all methods should be optimised in the same way.**
>
> **Our response:**
> - Indeed, we optimised all models in the same way.
>
> - We included hyper-parameters in the formulation just to keep equations in a very general form. But, in our implementation, we never used parameters defined in objective functions and aggregation, namely alpha, beta, gamma, eta in Equations 2, 3, 4 and 6 (i.e. they were all equal to 1 in our paper). We will remove these hyper-parameters in our update to make things clearer.
>
>  - Main hyper-parameters that we used are:
>
>    1.  Type of noise
>    2.  p , ratio of features corrupted in the subset,
>    3.  In the case of Gaussian noise, we used one parameter for variance (i.e. noise level).
>    4.  Number of subsets
>    5.  Overlap % between subsets (i.e. number of shared features between subsets)
>
> - We optimised all models by sweeping different noise types, percentage of features corrupted (i.e. p, or masking ratio), and noise level in the case of Gaussian noise. Thus, we treated all models equally. We used the same architecture since they were all autoencoders, and their main differences mostly come from how they augment the data.

---

> > ### Author Response · Authors · 2021-08-10
> > **Continuing our response to Reviewer TMPY (Part II)**
> >
> >
> > ---
> > > **8. The reviewer: Results for some baseline methods are taken from Yoon et al. “Extending the success of self-and semi-supervised learning to tabular domain”. Did the authors ensure that the train/validation/test splits are the same?**
> >
> > **Our response:**
> >
> > - We re-implemented VIME-self in Pytorch for 3 datasets to ensure that all methods are implemented in the same framework with same data pipeline and evaluated in the same way. We will update the paper with our own results to avoid any confusion and concern regarding differences in architecture, data pipeline and so on. Our result for VIME-self is:
> >
> >    |            	| MNIST          	|Income          	|Blog          	|
> >    |------------	|----------------	|----------------	|----------------	|
> >    | Vime-self: 	| 94.67 +/- 0.18	| 84.43 +/- 0.08  | 84.11 +/- 0.08. |
> >
> > - For the MNIST result above, we used the encoder architecture with [512-256-128], same as our base model for SubTab implementation. Moreover, we compared both methods using shallow encoder, in which Encoder has a single non-linear unit, since this is the original implementation of VIME-self for MNIST. The shallow encoder improved the performance of both methods as shown below:
> >
> >    |            	| MNIST          	|
> >    |------------	|----------------	|
> >    | Vime-self: 	| 96.67 +/- 0.13 	|
> >    | SubTab     	| 98.05 +/- 0.07 	|
> >
> >   Thus, it increased our SOTA score. We will also add this result to the paper with a short discussion in our update.
> >
> > - Finally, for MNIST, we used VIME’s dataloader to save the data as numpy file, and used their pre-processing steps such as scaling the data with 255, as it is shown in our git repo cited in the paper, to remove any concerns regarding differences in the data pre-processing. In summary, all models used the same data pipeline for all datasets to make sure that our comparison of the models is fair.
> >
> > ---
> > > **9. The reviewer: The untrained model achieves better than random performances. How did the authors initialise the weights? How sensitive is the initialisation? It would be helpful to provide the standard error in Figure 5 (and all figures in general).**
> >
> > **Our response:**
> > - We have not done anything specifically. We initialised using Pytorch’s default initialisation:
> >
> >    - stdv = 1. / math.sqrt(self.weight.size(1))
> >    - self.weight.data.uniform_(-stdv, stdv)
> >
> > - We can consider untrained model as a simple hash function. If the raw features of the data are informative enough, the untrained model will be able to map similar data points to close-by points in the latent space.
> > We did not use the standard errors in the figures for clarity, so we only plotted the mean. Also, Stdevs were small to be visible in the plots. But, we included them in the results in Table-1.
> >
> > - For untrained model in Figure-5, we re-ran the experiment for MNIST with 10 different random seeds, and below are the results. We checked the test accuracy using each of 7 subsets as it was done in the paper. As shown below, the variation is very small, and the results are not sensitive to the initialisation:
> >
> >      | SubSet   	| 1     	| 2     	| 3     	| 4     	| 5     	| 6     	| 7     	|
> >      |----------	|-------	|-------	|-------	|-------	|-------	|-------	|-------	|
> >      | Mean (%) 	| 22.13 	| 63.24 	| 71.79 	| 75.23 	| 69.20 	| 57.13 	| 35.11 	|
> >      | Stdev(%) 	| 0.09  	| 0.46  	| 0.59  	| 0.38  	| 0.34  	| 0.38  	| 0.24  	|
> >
> >    We will add this experiment to the supplementary material as a plot in our update.
> >
> >
> > ---
> > > **10. The reviewer: Intuitively, it seems that SubTab should work well with highly redundant datasets where several sets of features are informative of the underlying latent variables (e.g. MNIST or TCGA). How does the model perform when this is not the case? Do the claims from L49-L53 still hold?**
> >
> > **Our response:**
> > - To the best of our knowledge, the most commonly available tabular datasets fit the description of the reviewer i.e. highly redundant. But, we tried to include very diverse set of datasets in our experiments to measure our method’s generalizability.
> >
> > - As a follow-up to this question, we ran experiments on 3 synthetic datasets (that we generated using make_classification module of scikit-learn library [2]) .
> >
> > - **Datasets:**
> >      Each dataset has 10 classes, 10k samples, 10% of which is used as the test set. We generated them such that the clusters are not clearly separated (noisy boundary) to make the problem a little more difficult.
> >
> >   - **Specifics of datasets are:**
> >
> >
> >     | Dataset 	| Total Features 	| Informative Features 	| Redundant Features 	| Non-informative|
> >     |---------	|----------------	|----------------------	|--------------------	|------------------------------------------	|
> >     | 1       	| 1000           	| 12                   	| 30                 	| 958                                      	|
> >     | 2       	| 100            	| 60                   	| 30                 	| 10                                       	|
> >     | 3       	| 100            	| 4                    	| 30                 	| 66                                       	 |
> >
> >     *Redundant Features are linear combinations of informative ones.
> >
> >  - **SubTab set-up:**
> >     - We used SubTab with encoder architecture of [1024, 1024] hidden layers, of which the first one uses LeakyReLU, and second one is a linear layer.
> >     - We used 2 subsets with 25% overlap between them. Other parameters are p=0.2, gaussian noise with stdev=0.1.
> >     - We trained it using only reconstruction loss.
> >     - This set-up seems to work well for most tabular datasets as it did in other datasets reported in our paper, and we will add it as a guideline in the supplementary material.
> >
> >   - **Evaluation:**
> >     - We used Logistic regression to evaluate the test performance using the raw features of the data, and the embeddings from SubTab, and compared the test accuracy, and here are the results:
> >
> >       | Dataset 	| Raw Features (%) 	| SubTab embedding (%) 	|
> >       |---------	|------------------	|----------------------	|
> >       | 1       	| 31.2             	| 61.9                 	|
> >       | 2       	| 83.5             	| 90.5                 	|
> >       | 3       	| 79.9             	| 82.1                 	|
> >
> > So, we see that SubTab improved the results in all 3 datasets, as much as 100% in the most difficult dataset. We hope that this quick experiment gives a little bit more insight into SubTab method, but we caution the reviewer from drawing specific conclusions from that because there is more to tabular datasets than just the redundancy in our experience, and it needs to be evaluated in a case-by-case basis.
> >
> >
> >
> > ---
> > **References:**
> >
> > [1] https://proceedings.neurips.cc/paper/2020/file/7d97667a3e056acab9aaf653807b4a03-Supplemental.pdf
> >
> > [2] https://scikit-learn.org/stable/modules/generated/sklearn.datasets.make_classification.html

---

> > > ### Comment · Reviewer_TMPY · 2021-08-27
> > > **Thanks for the detailed response**
> > >
> > > The authors have adequately addressed my main concerns. I have increased my score from 6 to 7. Well done!

---

> > > > ### Author Response · Authors · 2021-08-27
> > > > **Thanks!**
> > > >
> > > > We truly appreciate the kind words. We thank the reviewer for the positive feedback and the vote of acceptance. We think that the paper got better with the feedback we received, and we are grateful to all the reviewers for that.

---

### Official Review · Reviewer_pA7F · 2021-07-16

**Rating:** 6
**Confidence:** 5

**Summary:**

This work presents a self-supervised method for the tabular domain, where structural information is not typically available (in contrast to images/text). The proposed method suggests partitioning input rows into subsets of features, where these subsets can have a certain overlap percentage as controlled by a hyperparameter and mapped to the latent space via an encoder where they define their reconstruction loss (authors provide two options 1—reconstructing the original subset from the subset latent representation 2—reconstructing the complete row from the subset representation), an optional contrastive loss, as well as a distance loss term in the projection embedding space where the negatives are chosen from subset latent representations from other rows. Orthogonal to their work, they leverage various noise strategies commonly used in the tabular domain, such as swap noise (imputing a feature from the same column from a different row), gaussian noise, and zero imputation. The authors compare with the previous SOTA method (VIME) on various tabular datasets.

**Ethics Review Area:**

["I don’t know"]

**Limitations And Societal Impact:**

Seems adequate.

**Main Review:**

Subsetting features (local view) to predict a global view is somewhat similar to random cropping and zooming in images, and I believe its application in the tabular domain is novel.

In section 2, as the authors used fully connected layers, by subsetting features, different columns will be fed to the same neuron at the input layer; this may lead to representations collapse to the same latent point. For example, consider a tabular data with six columns, and the rows 000101, 101000 then as per the method, these would map to similar latent representations if we consider subset features of length 3, but in fact, they may correspond to different manifolds when taken as a whole. This would also cause problems for the contrastive loss term/distance loss.

In section 2, l. 79, does random shuffling help performance? It is not clear how this could get rid of bias as a subset of features is being fed to the network, and from my understanding, ordering of subsets does not seem to play a role in the proposed method.

Section 2, l. 105 It would be great if authors could give intuition behind the contrastive/distance loss terms and why they make sense for their proposed augmentation

Section 3. How was the $\alpha_k$ in eqn. 3) set?

In subsection 3.1 it is unclear whether authors use the complete labelled training sets or treat only a subset of the training set as labelled. The ambiguity occurs when authors make a comparison with VIME[1], specifically, l. 196. VIME considers 10% of the training set as labelled and treats the rest as unlabeled in their results.

In subsection 3.3, the authors consider an [512-256-128] architecture, whereas VIME results use a single encoder layer of size 784 and 100-100 MLP with ReLU activation for their predictor, achieving 95.75 % or 94.55 % for VIME-self. There's confusion as authors claim to use the values from [1] when reporting results in Table 1, but in l.138 claim to use the same encoder size achieving lower accuracy values than the simpler architecture used in VIME. One may argue that this makes the comparison unfair/unclear. If authors tune architecture/hyperparameters for their method, the same should be done for other methods.

In table 1. Authors borrow values from VIME [1], and report the variant of VIME Self-SL only, is there a reason for this? VIME proposes an unsupervised pre-training step using swap noise (does not use labels), and enforces a consistency loss in their predictor (10% training labelled set) hence the semi-supervised. As a result, I believe their complete VIME method should be compared. Further, following my previous comments, it is unclear if the training setup is fair, specifically the architecture, and perhaps the preprocessing steps (using 1/255 normalization versus z normalization or min-max for MNIST may provide 1-3% accuracy gain in low capacity networks). Ideally, other methods should be implemented in the same framework with similar data pipelines and then compared. VIME MNIST results are reproducible and can be used to verify implementation errors.


L.204, the figure being referenced is the curve for the Blog dataset; I believe 5a) was the intended figure.
In this paragraph, does “train” refer to the pre-training step or training the predictor? In l. 214, If the encoder hasn’t been trained on the subsets, why can they be mapped to different points in the latent space? Following my previous example, it would seem representations for different rows could collapse to the same point. Why is subset 1 excluded (due to white pixels In MNIST?) ? Overall this paragraph is confusing.

There’s some minor inconsistencies e.g.  l. 227 (3, 4, 5) l.231 [3, 4, 5]


[1] Yoon, Jinsung, et al. "Vime: Extending the success of self-and semi-supervised learning to tabular domain." Advances in Neural Information Processing Systems 33 (2020).


**Time Spent Reviewing:**

5

---

> ### Author Response · Authors · 2021-08-10
> **Response to Reviewer pA7F (Part I)**
>
> We thank the reviewer for the insightful feedbacks and finding our work novel. Our response is below:
>
> ---
> >**1.  The reviewer considers our approach  novel since subsetting features (local view) to predict a global is view similar to cropping and zooming in images.**
>
> **Our response:**
>
> - Indeed, we get inspired by the methods such as cropping and zooming-in used in Computer Vision and tried to bring their success in Computer Vision to tabular domain.
>
>
>
> ---
> >**2.  The reviewer has a valid concern regarding the representation collapse and gives an example of a tabular data with six columns, and with the following two samples: 000101, 101000. The concern is that if we consider subset features of length 3, two samples would have the same subsets (000, 101), and they would map to similar latent representations even when they are supposed to be different. This issue would also cause problems for the contrastive loss term/distance loss.**
>
>
>
> **Our response:**
>
> Many thanks for this point. Indeed, we did have the same concerns, but we considered following points:
>
> 1. This is a very low risk in the case of tabular datasets since tabular data usually consists of heterogenous features (mixed features: categorical + continuous), which might have different statistical properties. It would be a bigger concern if the datasets consist of only small number of binary features (say only categorical data) or a MNIST-like dataset with much smaller feature dimension.
>
>
>
> 2.  We have two parameters to avoid sub-optimal solutions like the one described: “number of subsets”, and “overlap between subsets” (i.e. shared features between subsets)
>
>
>
> 3.  We introduced the idea of overlapping features (i.e. shared features) between subsets for two reasons:
>
>
>
>     **i)** To further reduce such possibility as described. For example, in the given example, when we have 30% overlap between subsets (i.e. 1 feature from each subset is shared), we would have
>
>     - 00**01** and **01**01 out of first sample 00**01**01 (‘01’ is shared)
>
>     - 10**10** and **10** 00 out of the second sample 10**10**00 (‘10’ is shared)
>
>     And this would reduce the risk.
>
>     **ii)**  To increase the correlation between subsets. This would also help with contrastive learning. However, in our experiments, we used contrastive loss only on MNIST and TCGA datasets since they were rich enough to take advantage of contrastive learning (i.e. it is more likely to have true negative samples in the batch if datasets have more classes in them).
>
> We will add this point to our paper either in the main section, or in the supplementary section.
>
> ---
> >**3.  The reviewer asked about how random shuffling mentioned in section 2, I. 79 helps performance, and thinks that ordering of subsets does not seem to play a role in the proposed method.**
>
> **Our response:**
>
> We agree with the reviewer, and the answer is no, random shuffling does not have any effect indeed. We will update the paper.
>
> ---
> >**4.  The reviewer asks for more intuition behind the contrastive/distance loss terms and why they make sense for their proposed augmentation, referring to Section 2, l. 105.**
>
> **Our response:**
>
> We get the inspiration from the literature in multi-view learning. We conjecture that the different views of the same data point should have similar representations, and hence close to each other in the latent space.
>
> We turn the task of learning from a single view of the data into a multi-view learning, in which each subset (i.e. sub-view) has different set of features, corresponding to different view of the same data point. And we wanted to bring different views of the data point closer to each other in the latent space by using contrastive and distance losses.
>
> - Via contrastive loss, we try to pull corresponding rows from all subsets closer to each other (different views of the data in the same row correspond to positive samples), while we use the rest of rows in the same batch as negatives of the positive pairs and pushed them away from the positives.
>
> - Via distance loss, we try to minimize the distance between different views of the same data point, again to bring them closer.
>
> However, as shown in our ablation study in Table-2 of the paper, we should emphasize that the gains from either loss were not substantial, but they were complementary. Real value in our gain came from two things:
>
> -  Learning representations by reconstructing all features from its subsets
>
> -  Aggregating the representations of subsets to build the representation of the sample (this is analogue to the aggregation of node representations in Graph Neural Networks, or pooling operation in Computer Vision).
>
> Finally, the contrastive loss is more applicable when the dataset is rich in number of classes or has a rich underlying structure. So, we used it only for MNIST and TCGA datasets.
>
> ---
>
> > **5.  The reviewer asks how αk in Equation 3 is set.**
>
> **Our response:**
>
> We just wanted to keep our formulation general in order to extend our method to an attention-based model in the feature. Alphas are always equal to 1 in this work. This is true for all other hyper-parameters defined in other equations as well (i.e. alpha, beta, gamma, eta…etc). We will remove them in our update to the paper since it causes a lot of confusion and gives the idea that our gains are coming from hyper-parameter tuning while, throughout the experiments, we kept them all 1’s in this work.
>
> ---
>
> >**6.  The reviewer asks whether we used labelled training sets since VIME uses 10% of the training set as labelled, referring to subsection 3.1 as well as to l. 196 in our paper.**
>
> **Our response:**
>
> We thank the reviewer and share the reviewer’s sentiment. To the best of our understanding, our evaluation is different than how VIME-self is being evaluated.
>
> **Our evaluation:**
>
> 1. We train our models using all the available data in the training set (i.e. after hyper-parameter tuning using validation set, we used all the available training data.)
>
> 2. Then, we get embeddings of both training and test set from the pre-trained model.
>
> 3. To evaluate the quality of the representation learned, we train a linear classifier (LogReg) using the embeddings of training set and test it using the embeddings of the test set.
>
> At the time of the submission, we thought that this is how VIME-self was evaluated and reported as well. We later realized that VIME-self is evaluated in the following way:
>
> **VIME-self:**
>
> 1. Train the model on 90% of the training set.
>
> 2. Get embeddings of 10% unused training set and all the test set.
>
> 3. Evaluate the representation quality by training a non-linear MLP model using the embeddings of the 10% training set and test the MLP on the embeddings of the test set.
>
> So, using the numbers from VIME paper is not appropriate in our case. Once we realised this, we evaluated VIME-self in a same procedure as our evaluation. We will update the revised paper by the new results of VIME-self for MNIST, Income, and Blog datasets. The results of VIME-self from the same evaluation are:
>
>    |            	| MNIST          	| Income         	| Blog           	|
>    |------------	|----------------	|----------------	|----------------	|
>    | Vime-self: 	| 94.67 +/- 0.18 	| 84.43 +/- 0.08 	| 84.11 +/- 0.27 	|
>
> ---
> > **7.  The reviewer brings up a good point, referring to Sub-section 3.3. The reviewer is concerned that we use [512-256-128] architecture for MNIST and that we re-use the numbers reported for MNIST in VIME, in which they use a single encoder layer of size 784. The concern is that this makes the comparison unfair/unclear and that architecture/hyperparameters tuning should be done for all methods in the same way for a fair comparison.**
>
> **Our response:**
>
> We agree with the reviewer. As mentioned before, we mis-understood how VIME-self was evaluated to begin with. Hence, as stated earlier, to remove any confusion, and unfairness, we re-implemented VIME-self in Pytorch. For MNIST, we used the same encoder architecture as SubTab, i.e. [512-256-128], and tuned the hyper-parameters of the model in the same way we did for SubTab. Swap-noise, same as in the VIME-self, is used for corrupting the data. Our MNIST result for VIME-self is 94.67 +/- 0.18%.
>
> Moreover, we compared both methods using shallow encoder as in the original Keras implementation of the VIME-self for MNIST. The shallow encoder improved the performance of both methods as shown below:
>
>    |            	| MNIST          	|
>    |------------	|----------------	|
>    | Vime-self: 	| 96.67 +/- 0.13 	|
>    | SubTab     	| **98.05 +/- 0.07**	|
>
> Thus, it increased our SOTA score. We will add this result to the paper with short discussion in our update.

---

> > ### Author Response · Authors · 2021-08-10
> > **Continuing our response to reviewer pA7F (Part II)**
> >
> >
> > ---
> > >**8.  The reviewer further asks questions about why we only compare our method to VIME Self-SL , and believes that we should compare against the complete VIME model, which uses the 10% of training set as labelled. The reviewer is also concerned that the training setup, architecture, and the differences in the pre-processing steps such as normalization might influence the results.**
> >
> > **Our response:**
> >
> > The reason why we compared to VIME-self only was that our work focuses on unsupervised / self-supervised representation learning only. Hence, semi-supervised approaches are out of scope of this submission.
> >
> > As stated earlier, we re-implemented VIME-self for 3 datasets to ensure that all methods are implemented and evaluated in the same framework with same data pipeline, and we will update the paper with the updated results to avoid any confusion and concern regarding differences in architecture, data pipeline, evaluation methods and so on.
> >
> > For MNIST, we used VIME’s dataloader to save the data as numpy file, and used their pre-processing steps such as scaling the data with 255, as it is shown in our git repo cited in the paper, to remove any concerns regarding differences in the data pre-processing. The data that we provided in our published git repo is saved directly from VIME’s dataloader.
> >
> > ---
> > >**9.  The reviewer points out a typo in L.204, and believes that Figure 5a was the intended figure.**
> >
> > **Our response:**
> >
> > Many thanks for this feedback. We were actually referring to the yellow line in Figure 3a. We will fix this typo in the updated paper.
> >
> > ---
> > >**10.  The reviewer points out to the word, “train” in L. 205, and is wondering whether it refers to the pre-training step or training the predictor**
> >
> > **Our response:**
> >
> > It refers to pre-training of the model. We will clarify it in our update to the paper.
> >
> > ---
> > >**11.  The reviewer points out to the l. 214 and wonder why the subsets can be mapped to the different points in the latent space, considering that the mode has not been trained. Referring to the earlier comments on the representation collapse, the reviewer thinks that the representations for different rows could collapse to the same point. Finally, the reviewer is asking why Subset-1 is excluded and finds the paragraph about the untrained model performance confusing.**
> >
> > **Our response:**
> >
> > We can consider untrained model as a simple hash function. If the raw features of the data are informative enough, the untrained model will be able to map similar data points to close-by points in the latent space. So, it can preserve some of the informativeness of the original features although it will be less than a trained model as shown in Figure-5.
> >
> > **About Subset 1:** Subset 1 is used as shown in Figure-5b and 5c. In Figure-5c, in x-axis, “all” refers to all subsets, including Subset-1 and 7.
> >
> > Experiments in Figure-5 is meant to show the informativeness of each individual subset as well as the informativeness of their aggregate representation before and after training the model. When aggregating the latent embeddings of subsets, we tried to show how their predictive power varies as we use different combination of subsets to get the aggregate representation.
> >
> > You are right that the Subset-1 and Subset-7 are very sparse (corresponding to mostly white pixels at top and bottom region of 28x28 images). Hence, their predictive power is very low compared to other subsets as shown in Figure-5b, which shows the predictive power of individual subsets. The most informative subsets are the ones corresponding to mid-region of the images (i.e. subsets 3, 4, 5).
> >
> >
> > **More about Figure-5:**
> >
> > We tried to measure the predictive power of each subset under different patterns of missingness during training and test time. The blue line with label “4” in Figure-5b corresponds to a case, in which we trained the model using only subset-4, meaning that subsets [1, 2, 3, 5, 6, 7] were not used during training, and assumed that we did not have access to them. Then, we evaluated the model trained on Subset-4 by using the embeddings of each of the other subsets to test their predictive power even though they were not used during training. As our baseline, we compared it to the embeddings of the subsets obtained from the untrained model.
> >
> > If we compare the predictive power of subset-3, by using its embeddings obtained from a model trained on subset-4 and the ones from an untrained model, we observe that the model trained on subset-4 still gives a more informative embeddings about subset-3.
> >
> > And we go on to repeat this experiment by using 4 other models trained on different combinations of the subsets: [4, 5], [3, 4, 5] and so on.
> >
> > We hope that this helps.
> >
> > ---
> >
> > >**12.  The reviewer points out to some minor inconsistencies e.g. l. 227 (3, 4, 5) versus l.231 [3, 4, 5]**
> >
> > **Our response:**
> >
> > Many thanks, this will be fixed in our update.

---

> ### Comment · Reviewer_pA7F · 2021-08-27
> **Thorough response**
>
> I am raising my score to a 6, as I believe authors have shown improvements over baselines and I appreciate the authors taking the time addressing the points raised through out the reviewing process.

---

> > ### Author Response · Authors · 2021-08-27
> > **Thanks!**
> >
> > We truly appreciate the reviewer's positive feedback, and thank the reviewer for the vote of acceptance.

---

### Official Review · Reviewer_2zYu · 2021-07-18

**Rating:** 6
**Confidence:** 5

**Summary:**

The paper proposes SubTab, a self-supervised (pre-training) method for learning good representations of tabular data. The method generates multiple views for each (unlabeled) example by selecting different (possibly overlapping) subsets of features. The model is trained with a three term loss, each averaged over all pairwise comparisons between the views -- a reconstruction term that teaches the model to reconstruct the full example from the cropped example, a contrastive term that encourages different crops of the same example to be closer together than crops of different examples (within the minibatch, infoNCE loss), and a distance term that encourages crops of the same example to be close together in L2 distance. Optionally, the input example can be corrupted by noise. After pre-training a linear classifier on the latent representation is trained using supervision. In this step, the latent embedding used is actually an average of those arising from feeding in several different random crops of the example. The authors test the method against VIME (Neurips 2020), context encoders, and denoising autoencoders on 5 datasets, which were used in the VIME paper.

**Ethical Concerns:**

No ethical concerns.

**Limitations And Societal Impact:**

Yes.

**Main Review:**

The paper has merits:
1) The motivation of the paper is good -- indeed, self-supervised learning for tabular data is relatively understudied.
2) The idea of generating views using random subsets of features is somewhat novel (though "Self-supervised Learning for Large-scale Item Recommendations" (https://arxiv.org/pdf/2007.12865.pdf) proposes using random feature masking for recommenders)
3) The authors compare against recent baselines and seem to outperform VIME.
4) Ablations are conducted.

However, there are concerns that push me to lean against acceptance:
1) More than just 5 datasets are needed to be convincing, especially given that both the datasets and model sizes are small so training should be quick (compared to say large language models). I would recommend benchmarking against 69 datasets from OpenML-CC18 -- as done in https://arxiv.org/abs/2106.15147
2) During test-time, the model applies a linear classifier on the average of the multiple latent embeddings (stemming from different crops of the original example). Averaging or "ensembling" in this way almost always boosts performance. If I understand correctly, this wasn't done for the baselines?
3) Even more baselines are needed. In particular, dropout -- both "feature dropout", where entries of the example are zeroed, and "model dropout", where dropout is applied to the intermediate neurons. Ensembling can be tested out here too. Also, I would be interested in seeing a comparison of SubTab with SCARF (https://arxiv.org/abs/2106.15147) -- though I understand this is concurrent work.
4) The method has quite a few hyper-parameters -- one for each term in the loss and also the type of noise to apply. Given that different datasets have different optimal hyper-parameters, it's not entirely clear that the gains over baselines aren't just a consequence of having more tunable hyper-parameters.

**Time Spent Reviewing:**

2

---

> ### Author Response · Authors · 2021-08-10
> **Response to Reviewer 2zYu (Part I)**
>
> We thank the reviewer for their insightful feedback and finding our method novel. Our response is below:
>
>
>
> ---
>
> > **1. The reviewer likes our idea of generating subsets using subset of features, and points to another recent work [4], since it proposes using random feature masking for recommenders.**
>
>
>
> #### Our response:
>
> - We would like to point out that the aforementioned paper’s method is not related to our method.
>
>
>
> - In our work, we generate multiple datasets (referred as “subsets” in the paper) with smaller feature dimension by assigning subset of features to them. This is similar to feature bagging in Random Forests, and analogue to cropping / zooming-in in Computer Vision. This is the first novelty  of our method. As a side benefit, this also has the advantage of reducing the number of parameters in the first layer of the model since each subset has a much smaller dimension than the original one, and parameter reduction might be substantial if the original data is very high dimensional as in the genomics datasets. Moreover, we can optionally add noise to a portion of each subset during training as we did in our paper.
> - Finally, we use an aggregation operation in tabular dataset domain, which is similar to the aggregation in Graph Neural Networks [1], and pooling operation in Computer Vision. This is the second novelty of our work.
>
> ---
>
> >**2. The reviewer is concerned that 5 datasets may not be enough, and suggests to benchmark our work using 69 datasets from OpenML-CC18, and refers to a concurrent paper [2].**
>
>
>
> #### Our response:
>
> - We would like note that we evaluated our method on the same benchmark datasets used by previous SOTA model, VIME-self, which was published in the last year’s NeurIPS. And we included two additional datasets to evaluate our method on a diverse set of datasets exemplar of commonly available data to show its general applicability.
>
>
>
> - We also believe that, given the time constraint and efforts needed, applying our method to 69 datasets to get aggregate results similar to the referred paper would deserve a separate paper, and out of the scope of our current paper as it would need detailed analysis on the results on each dataset. We are more than happy to do a follow-up paper on more general experiments in the future.
>
> ---
>
> >**3. The reviewer draws a similarity between our method of averaging multiple latent embeddings at test-time and the ensembling of models in ML literature. The reviewer also asks why we have not done the same thing for other models.**
>
>
>
> #### Our response:
>
> - Our method is not ensembling method, but is rather an analogue to mean-aggregation function used in Graph Neural Networks [1], in which a node’s representation is computed using aggregate of its neighbours. It is also similar to the mean-pooling operation commonly used in Computer Vision.
>
> - The novel idea behind our paper was to generate sub-views from the tabular dataset (i.e. subsets), and the aggregation of the latent representations of the sub-views was one of the natural results of our method and this is not applicable to other methods.
>
>
>
> - We also would like to make one more distinction that we use the aggregation to learn a representation rather than making a prediction. Ensemble learning is done using predictions of multiple models either explicitly (using either multiple different models, or using an ensemble model such as Random Forests), or implicitly (e.g. by using dropout):
>
>
>
>   - In ensembling by using multiple models, or using dropout method, the models are trained on the same set of features. And averaging is done at the output. In contrast, each subset in our method has different features, and aggregation is done at the input to get a representation before using it in a downstream task such as classification.
>
>
>
>   - In the case of Random Forest, it does use feature-bagging, similar to our subsets, but it can be considered as training multiple models trained on different subset of features of bootstrapped samples. However, again, the averaging in Random Forest is done by averaging the predictions of multiple sub-models (trees).
>
>
>
> - A more insight into what the aggregation is doing can be seen in Figure-6 in the Supplementary section of our paper, in which we show t-SNE plots of clusters of MNIST digits, and we compare the clusters obtained using 1 subset, the aggregate of 2 subsets and so on. As we aggregate more of the subsets, the overall representation is being improved.
>
>
>
> - Moreover, we ran an experiment using SubTab on MNIST by using different aggregation methods at test-time to prove our point, and here are the results (Please note that we also tried concatenating the latent embeddings of the subsets as shown):
>
>
>
>    | Aggregation method | Test Accuracy |
>    |--------------------  |---------------  |
>    | Mean | **97.87** |
>    | Sum  | 97.77 |
>    | Max  | 97.79 |
>    | Min  | 97.74 |
>    | Concatenation  | 95.92 |
>    |  | |
>
>
>
>    We will make sure that this main point about aggregation comes across clearly in our update to the paper.
>
> ---
>
> >**4. The reviewer suggested that we need more baselines and that we should try both “feature dropout", where entries of the example are zeroed, and "model dropout", where dropout is applied to the intermediate neurons.**
>
>
>
> #### Our response:
>
> - **Feature-dropout:** Experiments with feature dropout is already ran and mentioned in the paper. It corresponds to our experiments with zero-out noise on Denoising Autoencoder (DAE). We optimized the performance of all models by using 3 types of noise, and included the one that performed the best in our paper. In our experiments, zero-out noise always performed worse, and hence it is not included in the results. Table-1 in the paper shows which noise type performed best for each dataset.
>
>
>
> - **Model-dropout:** The focus of our paper is on the representation learning through data-augmentation in the context of autoencoder. That is why we compared our method against other methods that augments data through various means. Moreover, our work is orthogonal to any regularization method, including dropout. So, our method can be used together with dropout although we have not used it. Thus, we consider dropout method as complementary to our work rather than competitive method. However, we did run experiments for the rebuttal, using Autoencoder with dropout, using same architecture as the SubTab, and below are the results (Note that we used [512, 256, 512] encoder architecture for MNIST in this case):
>
>
>
>    | | |  |
>    |---------  |-----------------  |----------------  |
>    | **Dataset** |  **AE with Dropout**  |  **SubTab** |
>    | MNIST | 94.31 +/- 0.15  |  **97.86 +/- 0.07** |
>    | Income  | 85.00 +/- 0.11  |  **85.35 +/- 0.06** |
>    | Blog  | 84.18 +/- 0.20  |  **84.64 +/- 0.19** |
>    | Obesity | 62.74 +/- 4.38  |  **71.13 +/- 4.08** |
>    | TCGA  | 56.87 +/- 2.26  |  **58.25 +/- 1.36** |
>
>
>
>
> ---
>
> >**5. The reviewer asks us to compare our work to a concurrent work, SCARF [2]**.
>
>
>
> #### Our response:
>
> - Thanks for bringing this work (SCARF) to our attention. SCARF seems to be an application of SimCLR to tabular dataset. Moreover, SCARF corresponds to one of the instantiations of our method (i.e. our method is a more general form of contrastive approaches, in which one configuration would lead to SimCLR, or in this case SCARF). For example, we can use two copies of a single view (using all features) with added random noise (zero-out in the case of SCARF) to each copy and train the model using only contrastive loss. In fact, we have done this experiment for MNIST when we started this work, and it did not perform well (we obtained around ~94% test accuracy), compared to our current method, and so we did not include it in our discussions.
>
>
>
> - Moreover, a similar comparison (extension of SimCLR to tabular data) is discussed and experimented in the supplementary material of VIME, and they reported that it performed  similarly to their self-supervised version of VIME ( Table-5 in their NeurIPS supplementary material [3]).
>
>
>
> - Also, we should note that SCARF approach uses contrastive loss, which works well in the datasets that are rich in number of classes, or rich in structure since the probability of sampling false negatives would be low in this case. However, many of the tabular datasets are more appropriate for binary classification tasks, and they are not as rich as other data types such as images. As shown in Table-3 in our Supplementary material, we used contrastive loss only in MNIST and TCGA since they are rich enough to benefit from contrastive learning. Moreover, Figure-2 in SCARF paper shows very little to no gain in across all datasets, making our point.
>
>
>
> - Finally, as mentioned by the reviewer, the cited work is a concurrent work, which is submitted to arxiv one month after NeurIPS submission, June 29 to be exact. And it currently seems to be under the review. So, we feel that it would be unfair to ask us to compare our method to the cited work, and it does not comply with NeurIPS policy:
>
>
>
>     - Papers appearing less than two months before the submission deadline are generally considered concurrent to NeurIPS submissions.  Authors are not expected to compare to work that appeared only a month or two before the deadline.

---

> > ### Author Response · Authors · 2021-08-10
> > **Continuing our response to Reviewer 2zYu (Part II)**
> >
> >
> > ---
> >
> > >**6. The reviewer is worried that  our performance gains might be coming from hyper-parameter tuning since we have quite a few hyper-parameters**
> >
> >
> >
> > #### Our response:
> >
> > - Many thanks for bringing up this point of confusion. We included hyper-parameters in the formulation just to keep equations in a very general form. But, in our implementation, we never used parameters defined in objective functions and aggregation, namely alpha, beta, gamma, eta in equations 2, 3, 4 and 6 (i.e. they were all equal to 1).
> >
> >
> >
> > - As it can be seen from the ablation study (Table-2 in the paper), and results in Table-1, the main gains in performance mainly came from:
> >
> >
> >   - I.  Reconstruction of all features from its subsets
> >   - II.  Aggregation of the representations of subsets
> >
> >
> >
> > - Main hyper-parameters that we used are:
> >
> >    1.  Type of noise
> >
> >    2.  p , ratio of features corrupted in the subset,
> >
> >    3.  In the case of Gaussian noise, we used one parameter for variance (i.e. noise level).
> >
> >    4.  Number of subsets
> >
> >    5.  Overlap % between subsets (i.e. number of shared features between subsets)
> >
> >
> >
> > - Thus, compared to other methods, our method introduced only two hyper-parameters: “Number of subsets”, and “overlap between subsets”, which were the natural result of our proposal.
> >
> >
> >
> > - We optimised all models by sweeping different noise types, percentage of features corrupted (i.e. p, or masking ratio), and noise level in the case of Gaussian noise. Thus, all models are treated equally. We used the same architecture since they were all autoencoders, and their main differences mostly come from how they augment the data.
> >
> >   We will update the equations in the paper by removing those parameters and their normalisation factor (Z’s) to make it clear.
> >
> >
> >
> > ---
> >
> > **References:**
> >
> >
> >
> > [1] Semi-Supervised Classification with Graph Convolutional Networks: https://arxiv.org/pdf/1609.02907.pdf
> >
> > [2] SCARF  https://arxiv.org/abs/2106.15147
> >
> > [3] VIME supplementary material: https://proceedings.neurips.cc/paper/2020/file/7d97667a3e056acab9aaf653807b4a03-Supplemental.pdf
> >
> > [4] Self-supervised Learning for Large-scale Item Recommendations, https://arxiv.org/pdf/2007.12865.pdf

---

> ### Author Response · Authors · 2021-08-16
> **Follow-up: Response to Reviewer 2zYu (Part III)**
>
>
> We just wanted to do a follow-up on one of the request by the reviewer. The reviewer asked us whether we can do experiments on more data, such as the ones from OpenML-CC18 [1]. The reviewer further asked us whether we can compare our work to a concurrent work, SCARF [2].
>
> ---
> **Summary**
> -
> - **Running more experiments:**
> 	- Based on the reviewer's suggestions, we ran more experiments using 8 datasets from OpenML-CC18 [1].
>
> - **Comparison to SCARF [2]:**
> 	- We looked into comparing our results to that of SCARF [2]. However, we could not, because:
> 		- Their code is not available online (so we cannot replicate their work, considering the time line)
> 		- We could not do direct comparisons with SCARF[2] since they have reported their results in aggregate across 69 datasets using their "win" metric rather than test accuracy per each dataset.
>
>
> The details of our further experiments on 8 datasets is following:
>
> ---
> **Experiments**
> -
>
> - **Datasets:**
> 	-
>   - We believe that many of tabular datasets listed in OpenML-CC18 [1] is not suitable for representation learning because they either have very low number of features (e.g. < 10 features), or very low number of samples (e.g. < 1000 samples).
>
>   - Moreover, some datasets need more careful examination and time than the others since they might have some issues such as data leakage (including very informative features such as user IDs). So we had to be careful about which datasets we can use, given that we have limited time.
>
>   - Ideally, what we want for the task of representation learning is high-dimensional, large datasets with multiple-classes. The most tabular datasets do not fit this criteria, such as the ones shown in the table comparing 11 datasets below, namely:
>
> 	- Dataset 9: Diabetes,
> 	- Dataset 10: Blood transfusion service center
> 	- Dataset 11: Phoneme
>
>   - Thus, we excluded datasets such as these ones since they have a low number of features and/or a low number of samples, which make them unsuitable for using neural networks in general.
>
>   - Out of the 69 datasets, we picked 8 datasets shown in the table. Although Electricity dataset has only 8 features, we included it in our analysis since it does have relatively large sample size.
>
>
>   - **Data Pre-processing:** We cleaned up the datasets by removing rows with missing data if there is any, and/or by removing the features such as user ID.  We used MinMax scaling to scale all datasets, and split the data as 70-10-20% training, validation and test set as it is done in SCARF paper [2]. We trained the final models on 80% training set by combining training and validation set.
>
>   - **Summary of the datasets**: First 8 datasets shown in the table below are used for the experiments.
>
>     - **Table-1:** Datasets
>     | Dataset 	| Name                             	| Total Features 	| Number of Samples 	| Number of Classes 	|
>     |---------	|----------------------------------	|----------------	|-------------------	|-------------------	|
>     | **1**       	| **First Order Theorem Proving [3]**| 51             	| 6118              	| 6                 	|
>     | **2**       	| **Wall Robot [4]**                                | 24             	| 5456              	| 4                 	|
>     | **3**       	| **Gesture Phase Segmentation [5]**  | 32             	| 9873              	| 5                 	|
>     | **4**       	| **Ozone Level 8hr [6]**                  	| 72             	| 2534              	| 2                 	|
>     | **5**       	| **Electricity [7]**                      	| `8`              	| 45312             	| 2                 	|
>     | **6**       	| ***Texture [8]***                          	| 40             	| 5500              	| 11                	|
>     | **7**       	| ***DNA [9]***                              	| 180            	| 3186              	| 3                 	|
>     | **8**       	| ***Climate [10]***                          	| 20             	| 540               	| 2                 	|
>     | 9       	| Diabetes [11]                         	| `8`              	| 768               	| 2                 	|
>     | 10      	| Blood transfusion service center 	 [12]| `4`              	| 748               	| 2                 	|
>     | 11      	| Phoneme     [13]                      	| `5`              	| 5404              	| 2                 	|
>
>
>
> - **Models and Evaluation:**
> 	-
> 	- We trained and compared six models:
>
> 				1. Logistic Regression as our baseline
> 				2. Autoencoder
> 				3. Autoencoder with dropout (p=0.04)
> 				4. VIME-self
> 				5. SubTab
> 				6. SubTab with dropout (p=0.04)
>
> 	- All neural networks used the same four-layer encoder architecture used in SCARF paper [2]: [256, 256, 256, 256]
> 	- For the networks with drop-out, we used the same drop-out rate as SCARF [2], p=0.04
> 	- For neural networks, we trained and evaluated them with 10 different random seeds.
> 	- Evaluation is done by training a Logistic regression model using the embeddings of training set, and by testing it using the embeddings of the test set.
>
> - **Results:**
> 	-
>   - **Table-2:** Summary of the results
>   |   **Model**| **Dataset-1**| **Dataset-2**| **Dataset-3**| **Dataset-4**| **Dataset-5**| **Dataset-6**| **Dataset-7**| **Dataset-8**	|
>      |----------------------------------	|-----------------------------	|--------------	|-----------------------------------	|-----------------	|---------------	|--------------	|--------------	|--------------	|
>   | LogReg | 46.96 | 68.46| 46.93| 94.01| 76.09| **99.71**| **95.12**| **96.3**|
>   | Autoencoder|50.4+/-0.83| 86.83+/-0.91| 49.07+/-0.55| 94.84+/-0.34| 81.32+/-0.16| 99.34+/-0.28| 93.48+/-0.97| 95.01+/-0.9|
>   | Autoencoder with dropout| 50.52+/-0.71| 86.87+/-0.44| 49.43+/-1.17| 94.69+/-0.14| 81.54+/-0.36| 98.75+/-0.18 	| 91.48+/-0.43| 95.04+/-1.06|
>   | VIME-self| 44.99+/-0.9| 74.23+/-1.21| 46.08+/-0.37| 94.28+/-0.31|73.92+/-1.08| 95.49+/-0.88| 89.97+/-0.97| 95.56+/-0.42|
>   | **SubTab** | 50.8+/-0.76| 89.37+/-0.72|**50.33+/-0.86**| 94.74+/-0.28| 82.11+/-0.26| 99.59+/-0.22| 92.62+/-0.59 	| 93.89+/-1.55|
>   | **SubTab with dropout** |**51.48+/-0.77**|**89.81+/-0.69**| 49.93+/-0.77|**94.85+/-0.31**|**82.31+/-0.34**| 99.23+/-0.36| 91.41+/-1.03|93.33+/-0.77|
>
>
> - **Analysis:**
> 	-
> 	- Based on the results, we can make the following observations:
>
> 	  - SubTab outperforms VIME-self and Autoencoder approaches on the datasets [1-5]
>
> 	  - If the dataset is trivial (i.e. Logistic Regression already gives a very decent performance), the neural networks may not be needed, and the user might be better off using simple models such as Logistic Regression as this was the case in datasets [6, 7, 8, or  Texture, DNA,  and Climate datasets respectively].
>
> 	  - If the dataset is suitable for pre-training / representation learning, the SubTab tends to perform better than the other approaches, including SOTA model VIME-self.
>
> 	  - We will emphasise on the limitations of representation learning / pre-training on tabular datasets in our update to the paper as well as adding these results to the Supplementary section of our paper.
>
>
> - **Final thoughts:**
> 	-
> 	- There are couple of challenges in the tabular dataset domain. For example, there is not a publicly available, gold standard dataset in tabular domain (such as ImageNet in Computer Vision) that can be used for benchmarking models for various tasks such as representation learning.
>
>   - We think that one of the future directions in tabular domain should include the curation of ImageNet-like datasets for tabular domain.
>
> ---
> - **References:**
> 	-
> 	- [1] OpenML: https://www.openml.org/s/99
> 	- [2] SCARF: https://arxiv.org/pdf/2106.15147.pdf
> 	- [3] First Order Theorem Proving: https://www.openml.org/d/1475
> 	- [4] Wall Robot: https://www.openml.org/d/1497
> 	- [5] Gesture Phase Segmentation: https://www.openml.org/d/4538
> 	- [6] Ozone Level 8hr: https://www.openml.org/d/1487
> 	- [7] Electricity: https://www.openml.org/d/151
> 	- [8] Texture: https://www.openml.org/d/40499
> 	- [9] DNA: https://www.openml.org/d/40670
> 	- [10] Climate: https://www.openml.org/d/40918
> 	- [11] Diabetes: https://www.openml.org/d/37
> 	- [12] Blood transfusion service center: https://www.openml.org/d/1464
> 	- [13] Phoneme: https://www.openml.org/d/1489

---

> > ### Comment · Reviewer_2zYu · 2021-08-26
> > **RE: author response**
> >
> > In light of the detailed response, clarification on my concerns, and experiments from an additional handful of datasets from OpenML, I'll increase my score from 4 to 6.
> > With regards to comparing against SCARF: indeed, per policy, it is not obligatory to compare against concurrent papers. It was nice to see that you however did try to replicate the SCARF settings for the new experiments. Despite there not being open source code available, the method should be easy to implement (there's an algorithm block that spells it out), and if SubTab is accepted, I would encourage you all to compare against SCARF for the camera-ready.

---

> > > ### Author Response · Authors · 2021-08-27
> > > **Thanks!**
> > >
> > > We are grateful for the reviewer's positive feedback, and thank the reviewer for increasing the score. We will indeed consider replicating the SCARF if the time allows.

---

### Decision · Program_Chairs · 2021-09-27

**Decision:**

Accept (Poster)

**Comment:**

The authors have addressed most of the reviewer concerns during the review process, and I now suggest the acceptance of the paper. As the reviewers have acknowledged, the performance improvements are notable with an elegant and conceptually appealing method.

There is significant content in the author responses that need to be integrated into the paper prior to publication, especially on the new results on extra benchmark datasets, improved literature review, numerous clarification points on the method explanations, hyperparameter tuning, ablation studies, and description of limitations.